# Adaptive Conformal Inference Under Distribution Shift

**Isaac Gibbs**
Department of Statistics
Stanford University
igibbs@stanford.edu

**Emmanuel J. Candès**
Department of Statistics
Department of Mathematics
Stanford University
candes@stanford.edu

## Abstract

We develop methods for forming prediction sets in an online setting where the data generating distribution is allowed to vary over time in an unknown fashion. Our framework builds on ideas from conformal inference to provide a general wrapper that can be combined with any black box method that produces point predictions of the unseen label or estimated quantiles of its distribution. While previous conformal inference methods rely on the assumption that the data points are exchangeable, our adaptive approach provably achieves the desired coverage frequency over long-time intervals irrespective of the true data generating process. We accomplish this by modelling the distribution shift as a learning problem in a single parameter whose optimal value is varying over time and must be continuously re-estimated. We test our method, *adaptive conformal inference*, on two real world datasets and find that its predictions are robust to visible and significant distribution shifts.

## 1 Introduction

Machine learning algorithms are increasingly being employed in high stakes decision making processes. For instance, deep neural networks are currently being used in self-driving cars to detect nearby objects [2] and parole decisions are being made with the assistance of complex models that combine over a hundred features [1]. As the popularity of black box methods and the cost of making wrong decisions grow it is crucial that we develop tools to quantify the uncertainty of their predictions.

In this paper we develop methods for constructing prediction sets that are guaranteed to contain the target label with high probability. We focus specifically on an online learning setting in which we observe covariate-response pairs $\{(X_t, Y_t)\}_{t \in \mathbb{N}} \subseteq \mathbb{R}^d \times \mathbb{R}$ in a sequential fashion. At each time step $t \in \mathbb{N}$ we are tasked with using the previously observed data $\{(X_r, Y_r)\}_{1 \le r \le t-1}$ along with the new covariates, $X_t$, to form a prediction set $\hat{C}_t$ for $Y_t$. Then, given a target coverage level $\alpha \in (0, 1)$ our generic goal is to guarantee that $Y_t$ belongs to $\hat{C}_t$ at least $100(1 - \alpha)\%$ of the time.

Perhaps the most powerful and flexible tools for solving this problem come from conformal inference [see e.g. 34, 16, 32, 22, 31, 15, 3] . This framework provides a generic methodology for transforming the outputs of any black box prediction algorithm into a prediction set. The generality of this approach has facilitated the development of a large suite of conformal methods, each specialized to a specific prediction problem of interest [e.g. 30, 11, 23, 8, 24, 21]. With only minor exceptions all of these algorithms share the same common guarantee that if the training and test data are exchangeable, then the prediction set has valid marginal coverage $\mathbb{P}(Y_t \in \hat{C}_t) = 1 - \alpha$.

While exchangeability is a common assumption, there are many real-world applications in which we do not expect the marginal distribution of $(X_t, Y_t)$ to be stationary. For example, in finance and economics market behaviour can shift drastically in response to new legislation or major world

35th Conference on Neural Information Processing Systems (NeurIPS 2021).

events. Alternatively, the distribution of $(X_t, Y_t)$ may change as we deploy our prediction model in new environments. This paper develops *adaptive conformal inference* (ACI), a method for forming prediction sets that are robust to changes in the marginal distribution of the data. Our approach is both simple, in that it requires only the tracking of a single parameter that models the shift, and general as it can be combined with any modern machine learning algorithm that produces point predictions or estimated quantiles for the response. We show that over long time intervals ACI achieves the target coverage frequency without any assumptions on the data-generating distribution. Moreover, when the distribution shift is small and the prediction algorithm takes a certain simple form we show that ACI will additionally obtain approximate marginal coverage at most time steps.

## 1.1 Conformal inference

Suppose we are given a fitted regression model for predicting the value of $Y$ from $X$. Let $y$ be a candidate value for $Y_t$. To determine if $y$ is a reasonable estimate of $Y_t$, we define a conformity score $S(X, Y)$ that measures how well the value $y$ *conforms* with the predictions of our fitted model. For example, if our regression model produces point predictions $\hat{\mu}(X)$ then we could use a conformity score that measures the distance between $\hat{\mu}(X_t)$ and $y$. One such example is

$$S(X_t, y) = |\hat{\mu}(X_t) - y|.$$

Alternatively, suppose our regression model outputs estimates $\hat{q}(X; p)$ of the $p$th quantile of the distribution of $Y|X$. Then, we could use the method of conformal quantile regression (CQR) [28], which examines the signed distance between $y$ and fitted upper and lower quantiles through the score

$$S(X_t, y) = \max\{\hat{q}(X_t; \alpha/2) - y, y - \hat{q}(X_t; 1 - \alpha/2)\}.$$

Regardless of what conformity score is chosen the key issue is to determine how small $S(X_t, y)$ should be in order to accept $y$ as a reasonable prediction for $Y_t$. Assume we have a calibration set $\mathcal{D}_{\text{cal}} \subseteq \{(X_r, Y_r)\}_{1 \leq r \leq t-1}$ that is different from the data that was used to fit the regression model. Using this calibration set we define the fitted quantiles of the conformity scores to be

$$\hat{Q}(p) := \inf \left\{ s : \left( \frac{1}{|\mathcal{D}_{\text{cal}}|} \sum_{(X_r, Y_r) \in \mathcal{D}_{\text{cal}}} \mathbb{1}_{\{S(X_r, Y_r) \leq s\}} \right) \geq p \right\}, \tag{1}$$

and say that $y$ is a reasonable prediction for $Y_t$ if $S(X_t, y) \leq \hat{Q}(1 - \alpha)$.

The crucial observation is that if the data $\mathcal{D}_{\text{cal}} \cup \{(X_t, Y_t)\}$ are exchangeable and we break ties uniformly at random then the rank of $S(X_t, Y_t)$ amongst the points $\{S(X_r, Y_r)\}_{(X_r, Y_r) \in \mathcal{D}_{\text{cal}}} \cup \{S(X_t, Y_t)\}$ will be uniform. Therefore,

$$\mathbb{P}(S(X_t, Y_t) \leq \hat{Q}(1 - \alpha)) = \frac{\lceil |\mathcal{D}_{\text{cal}}|(1 - \alpha) \rceil}{|\mathcal{D}_{\text{cal}}| + 1}.$$

Thus, defining our prediction set to be $\hat{C}_t := \{y : S(X_t, y) \leq \hat{Q}(1 - \alpha)\}$ gives the marginal coverage guarantee

$$\mathbb{P}(Y_t \in \hat{C}_t) = \mathbb{P}(S(X_t, Y_t) \leq \hat{Q}(1 - \alpha)) = \frac{\lceil |\mathcal{D}_{\text{cal}}|(1 - \alpha) \rceil}{|\mathcal{D}_{\text{cal}}| + 1}.$$

By introducing additional randomization this generic procedure can be altered slightly to produce a set $\hat{C}_t$ that satisfies the exact marginal coverage guarantee $\mathbb{P}(Y_t \in \hat{C}_t) = 1 - \alpha$ [34]. For the purposes of this paper this adjustment is not critical and so we omit the details here. Additionally, we remark that the method outlined above is often referred to as *split* or *inductive* conformal inference [27, 34, 26]. This refers to the fact that we have split the observed data between a training set used to fit the regression model and a withheld calibration set. The adaptive conformal inference method developed in this article can also be easily adjusted to work with full conformal inference in which data splitting is avoided at the cost of greater computational resources [34].

## 2 Adapting conformal inference to distribution shifts

Up until this point we have been working with a single score function $S(\cdot)$ and quantile function $\hat{Q}(\cdot)$. In the general case where the distribution of the data is shifting over time both these functions should

be regularly re-estimated to align with the most recent observations. Therefore, we assume that at each time $t$ we are given a fitted score function $S_t(\cdot)$ and corresponding quantile function $\hat{Q}_t(\cdot)$. We define the realized miscoverage rate of the prediction set $\hat{C}_t(\alpha) := \{y : S_t(X_t, y) \leq \hat{Q}_t(1-\alpha)\}$ as

$$M_t(\alpha) := \mathbb{P}(S_t(X_t, Y_t) > \hat{Q}_t(1-\alpha)),$$

where the probability is over the test point $(X_t, Y_t)$ as well as the data used to fit $S_t(\cdot)$ and $\hat{Q}_t(\cdot)$.

Now, since the distribution generating the data is non-stationary we do not expect $M_t(\alpha)$ to be equal, or even close to, $\alpha$. Even so, we can still postulate that if the conformity scores used to fit $\hat{Q}_t(\cdot)$ cover the bulk of the distribution of $S_t(X_t, Y_t)$ then there may be an alternative value $\alpha_t^* \in [0, 1]$ such that $M_t(\alpha_t^*) \cong \alpha$. More rigorously, assume that with probability one, $\hat{Q}_t(\cdot)$ is continuous, non-decreasing and such that $\hat{Q}_t(0) = -\infty$ and $\hat{Q}_t(1) = \infty$. This does not hold for the split conformal quantile functions defined in (1), but in the case where there are no ties amongst the conformity scores we can adjust our definition to guarantee this by smoothing over the jump discontinuities in $\hat{Q}(\cdot)$. Then, $M_t(\cdot)$ will be non-decreasing on $[0, 1]$ with $M_t(0) = 0$ and $M_t(1) = 1$ and so we may define

$$\alpha_t^* := \sup\{\beta \in [0, 1] : M_t(\beta) \leq \alpha\}.$$

Moreover, if we additionally assume that

$$\mathbb{P}(S_t(X_t, Y_t) = \hat{Q}_t(1 - \alpha_t^*)) = 0,$$

then we will have that $M_t(\alpha_t^*) = \alpha$. So, in particular we find that by correctly calibrating the argument to $\hat{Q}_t(\cdot)$ we can achieve either approximate or exact marginal coverage.

To perform this calibration we will use a simple online update. This update proceeds by examining the empirical miscoverage frequency of the previous prediction sets and then decreasing (resp. increasing) our estimate of $\alpha_t^*$ if the prediction sets were historically under-covering (resp. over-covering) $Y_t$. In particular, let $\alpha_1$ denote our initial estimate (in our experiments we will choose $\alpha_1 = \alpha$). Recursively define the sequence of miscoverage events

$$\text{err}_t := \begin{cases} 1, & \text{if } Y_t \notin \hat{C}_t(\alpha_t), \\ 0, & \text{otherwise,} \end{cases} \quad \text{where } \hat{C}_t(\alpha_t) := \{y : S_t(X_t, y) \leq \hat{Q}_t(1 - \alpha_t)\}.$$

Then, fixing a step size parameter $\gamma > 0$ we consider the simple online update

$$\alpha_{t+1} := \alpha_t + \gamma(\alpha - \text{err}_t). \tag{2}$$

We refer to this algorithm as *adaptive conformal inference*. Here, $\text{err}_t$ plays the role of our estimate of the historical miscoverage frequency. A natural alternative to this is the update

$$\alpha_{t+1} = \alpha_t + \gamma\left(\alpha - \sum_{s=1}^{t} w_s \text{err}_s\right), \tag{3}$$

where $\{w_s\}_{1 \leq s \leq t} \subseteq [0, 1]$ is a sequence of increasing weights with $\sum_{s=1}^{t} w_s = 1$. This update has the appeal of more directly evaluating the recent empirical miscoverage frequency when deciding whether or not to lower or raise $\alpha_t$. In practice, we find that (2) and (3) produce almost identical results. For example, in Section A.3 in the Appendix we show some sample trajectories for $\alpha_t$ obtained using the update (3) with

$$w_s := \frac{0.95^{t-s}}{\sum_{s'=1}^{t} 0.95^{t-s'}}.$$

We find that these trajectories are very similar to those produced by (2). The main difference is that the trajectories obtained with (3) are smoother with less local variation in $\alpha_t$. In the remainder of this article we will focus on (2) for simplicity.

## 2.1 Choosing the step size

The choice of $\gamma$ gives a tradeoff between adaptability and stability. While raising the value of $\gamma$ will make the method more adaptive to observed distribution shifts, it will also induce greater volatility in

the value of $\alpha_t$. In practice, large fluctuations in $\alpha_t$ may be undesirable as it allows the method to oscillate between outputting small conservative and large anti-conservative prediction sets.

In Theorem 4.2 we give an upper bound on $(M_t(\alpha_t) - \alpha)^2$ that is optimized by choosing $\gamma$ proportional to $\sqrt{|\alpha_{t+1}^* - \alpha_t^*|}$. While not directly applicable in practice, this result supports the intuition that in environments with greater distributional shift the algorithm needs to be more adaptable and thus $\gamma$ should be chosen to be larger. In our experiments we will take $\gamma = 0.005$. This value was chosen because it was found to give relatively stable trajectories for $\alpha_t$ while still being sufficiently large as to allow $\alpha_t$ to adapt to observed shifts. In agreement with the general principles outlined above we found that larger values of $\gamma$ also successfully protect against distribution shifts, while taking $\gamma$ to be too small causes adaptive conformal inference to perform similar to non-adaptive methods that hold $\alpha_t = \alpha$ constant across time.

## 2.2  Real data example: predicting market volatility

We apply ACI to the prediction of market volatility. Let $\{P_t\}_{1 \le t \le T}$ denote a sequence of daily open prices for a stock. For all $t \ge 2$, define the return $R_t := (P_t - P_{t-1})/P_{t-1}$ and realized volatility $V_t = R_t^2$. Our goal is to use the previously observed returns $X_t := \{R_s\}_{1 \le s \le t-1}$ to form prediction sets for $Y_t := V_t$. More sophisticated financial models might augment $X_t$ with additional market covariates (available to the analyst at time $t-1$). As the primary purpose of this section is to illustrate adaptive conformal inference we work with only a simple prediction method.

We start off by forming point predictions using a GARCH(1,1) model [4]. This method assumes that $R_t = \sigma_t \epsilon_t$ with $\epsilon_2, \ldots, \epsilon_T$ taken to be i.i.d. $\mathcal{N}(0,1)$ and $\sigma_t$ satisfying the recursive update

$$\sigma_t^2 = \omega + \tau V_{t-1} + \beta \sigma_{t-1}^2.$$

This is a common approach used for forecasting volatility in economics. In practice, shifting market dynamics can cause the predictions of this model to become inaccurate over large time periods. Thus, when forming point predictions we fit the model using only the last 1250 trading days (i.e. approximately 5 years) of market data. More precisely, for all times $t > 1250$ we fit the coefficients $\hat{\omega}_t, \hat{\tau}_t, \hat{\beta}_t$ as well as the sequence of variances $\{\hat{\sigma}_s^t\}_{1 \le s \le t-1}$ using only the data $\{R_r\}_{t-1250 \le r < t}$. Then, our point prediction for the realized volatility at time $t$ is

$$(\hat{\sigma}_t^t)^2 := \hat{\omega}_t + \hat{\tau}_t V_{t-1} + \hat{\beta}_t (\hat{\sigma}_{t-1}^t)^2.$$

To form prediction intervals we define the sequence of conformity scores

$$S_t := \frac{|V_t - (\hat{\sigma}_t^t)^2|}{(\hat{\sigma}_t^t)^2}$$

and the corresponding quantile function

$$\hat{Q}_t(p) := \inf \left\{ x : \frac{1}{1250} \sum_{r=t-1250}^{t-1} \mathbb{1}_{S_r \le x} \ge p \right\}.$$

Then, our prediction set at time $t$ is

$$\hat{C}_t(\alpha_t) := \left\{ v : \frac{|v - (\hat{\sigma}_t^t)^2|}{(\hat{\sigma}_t^t)^2} \le \hat{Q}_t(1 - \alpha_t) \right\},$$

where $\{\alpha_t\}$ is initialized with $\alpha_{1250} = \alpha = 0.1$ and then updated recursively as in (2).

We compare this algorithm to a non-adaptive alternative that takes $\alpha_t = \alpha$ fixed. To measure the performance of these methods across time we examine their local coverage frequencies defined as the average coverage rate over the most recent two years, i.e.

$$\text{localCov}_t := 1 - \frac{1}{500} \sum_{r=t-250+1}^{t+250} \text{err}_r. \tag{4}$$

If the methods perform well then we expect the local coverage frequency to stay near the target value $1 - \alpha$ across all time points.

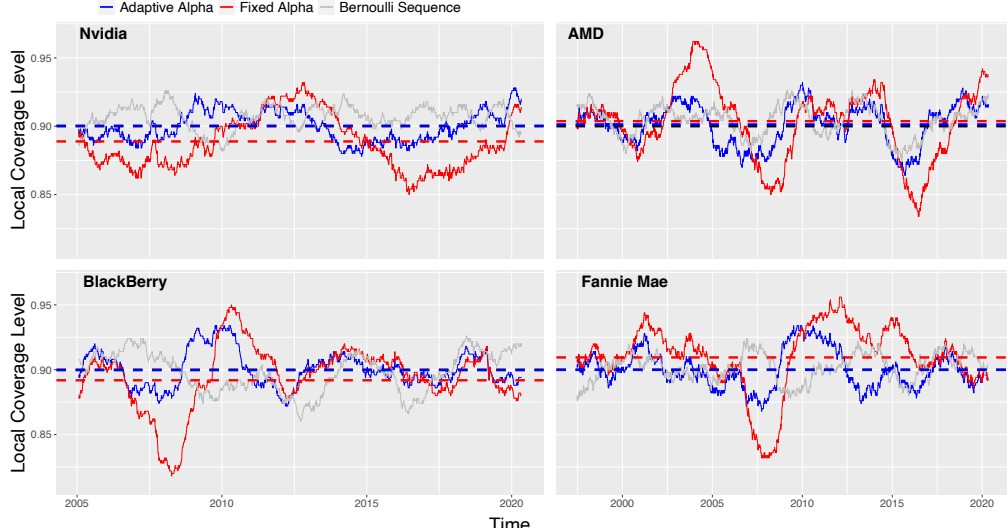

Figure 1: Local coverage frequencies for adaptive conformal (blue), a non-adaptive method that holds $\alpha_t = \alpha$ fixed (red), and an i.i.d. Bernoulli(0.1) sequence (grey) for the prediction of stock market volatility. The coloured dotted lines mark the average coverage obtained across all time points, while the black line indicates the target level of $1 - \alpha = 0.9$.

Daily open prices were obtained from publicly available datasets published by *The Wall Street Journal*. The realized local coverage frequencies for the non-adaptive and adaptive conformal methods on four different stocks are shown in Figure 1. These stocks were selected out of a total of 12 stocks that we examined because they showed a clear failure of the non-adaptive method. Adaptive conformal inference was found to perform well in all cases (see Figure 9 in the appendix).

As a visual comparator, the grey curves show the moving average $1 - \frac{1}{500} \sum_{r=t-250+1}^{t+250} I_r$ for sequences $\{I_t\}_{1 \le t \le T}$ that are i.i.d. Bernoulli(0.1). We see that the local coverage frequencies obtained by adaptive conformal inference (blue lines) always stay within the variation that would be expected from an i.i.d. Bernoulli sequence. On the other hand, the non-adaptive method undergoes large excursions away from the target level of $1 - \alpha = 0.9$ (red lines). For example, in the bottom right panel we can see that the non-adaptive method fails to cover the realized volatility of Fannie Mae during the 2008 financial crisis, while the adaptive method is robust to this event (see Figure 4 in the Appendix for a plot of the price of Fannie Mae over this time period).

## 3 Related Work

Prior work on conformal inference has considered two different types of distribution shift [33, 10]. In both cases the focus was on environments in which the calibration data is drawn i.i.d. from a single distribution $P_0$, while the test point comes from a second distribution $P_1$. In this setting Tibshirani et al. [33] showed that valid prediction sets can be obtained by re-weighting the calibration data using the likelihood ratio between $P_1$ and $P_0$. However, this requires the conditional distribution of $Y|X$ to be constant between training and testing and the likelihood ratio $P_1(X)/P_0(X)$ to be either known or very accurately estimated. On the other hand, Cauchois et al. [10] develop methods for forming prediction sets that are valid whenever $P_1$ and $P_0$ are close in $f$-divergence. Similar to our work, they show that if $D_f(P_1||P_0) \le \rho$ then there exists a conservative value $\alpha_\rho \in (0, 1)$ such that

$$M(\alpha_\rho) := \mathbb{P}(S(X_t, Y_t) > \hat{Q}(1 - \alpha_\rho)) \le \alpha.$$

The difference between our approach and theirs is twofold. First, while they fix a single conservative value $\alpha_\rho$ our methods aim to estimate the optimal choice $\alpha^*$ satisfying $M(\alpha^*) = \alpha$. This is not possible in the setting of [10] as they do not observe any data from which the size of the distribution shift can be estimated. Second, while they consider only one training and one testing distribution we work in a fully online setting in which the distribution is allowed to shift continuously over time.

## 4 Coverage guarantees

### 4.1 Distribution-free results

In this section we outline the theoretical coverage guarantees of adaptive conformal inference. We will assume throughout that with probability one $\alpha_1 \in [0, 1]$ and $\hat{Q}_t$ is non-decreasing with $\hat{Q}_t(x) = -\infty$ for all $x < 0$ and $\hat{Q}_t(x) = \infty$ for all $x > 1$. Our first result shows that over long time intervals adaptive conformal inference obtains the correct coverage frequency irrespective of any assumptions on the data-generating distribution.

**Lemma 4.1** *With probability one we have that $\forall t \in \mathbb{N}$, $\alpha_t \in [-\gamma, 1 + \gamma]$.*

**Proof:** Assume by contradiction that with positive probability $\{\alpha_t\}_{t \in \mathbb{N}}$ is such that $\inf_t \alpha_t < -\gamma$ (the case where $\sup_t \alpha_t > 1 + \gamma$ is identical). Note that $\sup_t |\alpha_{t+1} - \alpha_t| = \sup_t \gamma |\alpha - \mathrm{err}_t| < \gamma$. Thus, with positive probability we may find $t \in \mathbb{N}$ such that $\alpha_t < 0$ and $\alpha_{t+1} < \alpha_t$. However,

$$\alpha_t < 0 \implies \hat{Q}_t(1 - \alpha_t) = \infty \implies \mathrm{err}_t = 0 \implies \alpha_{t+1} = \alpha_t + \gamma(\alpha - \mathrm{err}_t) \geq \alpha_t$$

and thus $\mathbb{P}(\exists t \text{ such that } \alpha_{t+1} < \alpha_t < 0) = 0$. We have reached a contradiction. $\qquad\square$

**Proposition 4.1** *With probability one we have that for all $T \in \mathbb{N}$,*

$$\left| \frac{1}{T} \sum_{t=1}^{T} err_t - \alpha \right| \leq \frac{\max\{\alpha_1, 1 - \alpha_1\} + \gamma}{T\gamma}. \tag{5}$$

*In particular,* $\lim_{T \to \infty} \frac{1}{T} \sum_{t=1}^{T} err_t \overset{a.s.}{=} \alpha$.

**Proof:** By expanding the recursion defined in (2) and applying Lemma 4.1 we find that

$$[-\gamma, 1 + \gamma] \ni \alpha_{T+1} = \alpha_1 + \sum_{t=1}^{T} \gamma(\alpha - \mathrm{err}_t).$$

Rearranging this gives the result. $\qquad\square$

Proposition 4.1 puts no constraints on the data generating distribution. One may immediately ask whether these results can be improved by making mild assumptions on the distribution shifts. We argue that without assumptions on the quality of the initialization the answer to this question is negative. To understand this, consider a setting in which there is a single fixed optimal target $\alpha^* \in [0, 1]$ and assume that

$$M_t(p) = M(p) = \begin{cases} \alpha + \frac{1-\alpha}{1-\alpha^*}(p - \alpha^*), & \text{if } p > \alpha^*, \\ \alpha + \frac{\alpha}{\alpha^*}(p - \alpha^*) & \text{if } p \leq \alpha^*. \end{cases}.$$

Suppose additionally that $\mathbb{E}[\mathrm{err}_t | \alpha_t] = M(\alpha_t)$.[1] In order to simplify the calculations consider the noiseless update $\alpha_{t+1} = \alpha_t + \gamma(\alpha - M(\alpha_t)) = \alpha_t + \gamma(\alpha - \mathbb{E}[\mathrm{err}_t | \alpha_t])$. Intuitively, the noiseless update can be viewed as the average case behaviour of (2). Now, for any initialization $\alpha_1$ and any $\gamma \leq \min\{\frac{1-\alpha^*}{1-\alpha}, \frac{\alpha^*}{\alpha}\}$ there exists a constant $c \in \{\frac{1-\alpha}{1-\alpha^*}, \frac{\alpha}{\alpha^*}\}$ such that for all $t$, $M(\alpha_t) - \alpha = c(\alpha_t - \alpha^*)$. So, we have that

$$\mathbb{E}[\mathrm{err}_t] - \alpha = c\mathbb{E}[\alpha_t - \alpha^*] = c\mathbb{E}[\alpha_{t-1} + \gamma(\alpha - M_{t-1}(\alpha_{t-1})) - \alpha^*] = c(1 - c\gamma)\mathbb{E}[\alpha_{t-1} - \alpha^*].$$

Repeating this calculation recursively gives that

$$\mathbb{E}[\mathrm{err}_t] - \alpha = c(1 - c\gamma)^{t-1}\mathbb{E}[\alpha_1 - \alpha^*] = c(1 - c\gamma)^{t-1}(\alpha_1 - \alpha^*),$$

and thus,

$$\left| \frac{1}{T} \sum_{t=1}^{T} \mathbb{E}[\mathrm{err}_t] - \alpha \right| = \frac{1 - (1 - c\gamma)^T}{T\gamma} |\alpha_1 - \alpha^*|.$$

The comparison of this bound to (5) is self-evident. The main difference is that we have replaced $\max\{1 - \alpha_1, \alpha_1\}$ with $|\alpha_1 - \alpha^*|$. This arises from the fact that $\alpha^* \in (0, 1)$ is arbitrary and thus $\max\{1 - \alpha_1, \alpha_1\}$ is the best possible upper bound on $|\alpha_1 - \alpha^*|$. So, we view Proposition 4.1 as both an agnostic guarantee that shows that our method gives the correct long-term empirical coverage frequency irrespective of the true data generating process, and as an approximately tight bound on the worst-case behaviour immediately after initialization.

---

[1]This last assumption is in general only true if $\hat{Q}_t(\cdot)$ and $\alpha_t$ are fit independently of one another.

## 4.2 Performance in a hidden Markov model

Although we believe Proposition 4.1 is an approximately tight characterization of the behaviour after initialization, we can still ask whether better bounds can be obtained for large time steps. In this section we answer this question positively by showing that if $\alpha_1$ is initialized appropriately and the distribution shift is small, then tighter coverage guarantees can be given. In order to obtain useful results we will make some simplifying assumptions about the data generating process. While we do not expect these assumptions to hold exactly in any real-world setting, we do consider our results to be representative of the true behaviour of adaptive conformal inference and we expect similar results to hold under alternative models.

### 4.2.1 Setting

We model the data as coming from a hidden Markov model. In particular, we let $\{A_t\}_{t \in \mathbb{N}} \subseteq \mathcal{A}$ denote the underlying Markov chain for the environment and we assume that conditional on $\{A_t\}_{t \in \mathbb{N}}$, $\{(X_t, Y_t)\}_{t \in \mathbb{N}}$ is an independent sequence with $(X_t, Y_t) \sim P_{A_t}$ for some collection of distributions $\{P_a : a \in \mathcal{A}\}$. In order to simplify our calculations, we assume additionally that the estimated quantile function $\hat{Q}_t(\cdot)$ and score function $S_t(\cdot)$ do not depend on $t$ and we denote them by $\hat{Q}(\cdot)$ and $S(\cdot)$. This occurs for example in the split conformal setting with fixed training and calibration sets.

In this setting, $\{(\alpha_t, A_t)\}_{t \in \mathbb{N}}$ forms a Markov chain on $[-\gamma, 1 + \gamma] \times \mathcal{A}$. We assume that this chain has a unique stationary distribution $\pi$ and that $(\alpha_1, A_1) \sim \pi$. This implies that $(\alpha_t, A_t, \text{err}_t)$ is a stationary process and thus will greatly simplify our characterization of the behaviour of $\text{err}_t$. While there is little doubt that the theory can be extended, recall our that main goal is to get useful and simple results. That said, what we really have in mind here is that $\{A_t\}_{t \in \mathbb{N}}$ is sufficiently well-behaved to guarantee that $(\alpha_t, A_t)$ has a limiting stationary distribution. In Section A.5 we give an example where this is indeed provably the case. Lastly, the assumption that $(\alpha_1, A_1) \sim \pi$ is essentially equivalent to assuming that we have been running the algorithm for long enough to exit the initialization phase described in Section 4.1.

### 4.2.2 Large deviation bound for the errors

Our first observation is that $\text{err}_t$ has the correct average value. More precisely, by Proposition 4.1 we have that $\lim_{T \to \infty} T^{-1} \sum_{t=1}^{T} \text{err}_t \overset{a.s.}{=} \alpha$ and since $\text{err}_t$ is stationary it follows that $\mathbb{E}[\text{err}_t] = \alpha$. Thus, to understand the deviation of $T^{-1} \sum_{t=1}^{T} \text{err}_t$ from $\alpha$ we simply need to characterize the dependence structure of $\{\text{err}_t\}_{t \in \mathbb{N}}$.

We accomplish this in Theorem 4.1, which gives a large deviation bound on $|T^{-1} \sum_{t=1}^{T} \text{err}_t - \alpha|$. The idea behind this result is to decompose the dependence in $\{\text{err}_t\}_{t \in \mathbb{N}}$ into two parts. First, there is dependence due to the fact that $\alpha_t$ is a function of $\{\text{err}_r\}_{1 \leq r \leq t-1}$. In Section A.7 in the Appendix we argue that this dependence induces a negative correlation and thus the errors concentrate around their expectation at a rate no slower than that of an i.i.d. Bernoulli sequence. This gives rise to the first term in (6), which is what would be obtained by applying Hoeffding's inequality to an i.i.d. sequence. Second, there is dependence due to the fact that $A_t$ depends on $A_{t-1}$. More specifically, consider a setting in which the distribution of $Y|X$ has more variability in some states than others. The goal of adaptive conformal inference is to adapt to the level of variability and thus return larger prediction sets in states where the distribution of $Y|X$ is more spread. However, this algorithm is not perfect and as a result there may be some states $a \in \mathcal{A}$ in which $\mathbb{E}[\text{err}_t | A_t = a]$ is biased away from $\alpha$. Furthermore, if the environment tends to spend long stretches of time in more variable (or less variable) states this will induce a positive dependence in the errors and cause $T^{-1} \sum_{t=1}^{T} \text{err}_t$ to deviate from $\alpha$. To control this dependence we use a Bernstein inequality for Markov chains to bound $|T^{-1} \sum_{t=1}^{T} \mathbb{E}[\text{err}_t | A_t] - \alpha|$. This gives rise to the second term in (6).

**Theorem 4.1** *Assume that $\{A_t\}_{t \in \mathbb{N}}$ has non-zero absolute spectral gap $1 - \eta > 0$. Let*

$$B := \sup_{a \in \mathcal{A}} |\mathbb{E}[\text{err}_t | A_t = a] - \alpha| \quad \text{and} \quad \sigma_B^2 := \mathbb{E}[(\mathbb{E}[\text{err}_t | A_t] - \alpha)^2].$$

*Then,*

$$\mathbb{P}\left( \left| \frac{1}{T} \sum_{t=1}^{T} \text{err}_t - \alpha \right| \geq \epsilon \right) \leq 2 \exp\left( -\frac{T\epsilon^2}{8} \right) + 2 \exp\left( -\frac{T(1 - \eta)\epsilon^2}{8(1 + \eta)\sigma_B^2 + 20 B \epsilon} \right). \tag{6}$$

A formal proof of this result can be found in Section A.7. The quality of this concentration inequality will depend critically on the size of the bias terms $B$ and $\sigma_B^2$. Before proceeding, it is important that we emphasize that the definitions of $B$ and $\sigma_B^2$ are independent of the choice of $t$ owing to the fact that $(\alpha_t, A_t, \text{err}_t)$ is assumed stationary. Now, to understand these quantities, let

$$M(p|a) := \mathbb{P}(S(X_t, Y_t) > \hat{Q}(1-p)|A_t = a)$$

denote the realized miscoverage level in state $a \in \mathcal{A}$ obtained by the quantile $\hat{Q}(1-p)$. Assume that $M(p|a)$ is continuous. This will happen for example when $\hat{Q}(\cdot)$ is continuous and $S(X_t, Y_t)|A_t = a$ is continuously distributed. Then, there exists an optimal value $\alpha_a^*$ such that $M(\alpha_a^*|a) = \alpha$. Lemma A.4 in the Appendix shows that if in addition $M(\cdot|a)$ admits a second order Taylor expansion, then

$$B \leq C\left(\gamma + \gamma^{-1}\sup_{a\in\mathcal{A}}\sup_{k\in\mathbb{N}}\mathbb{E}[|\alpha_{A_{t+1}}^* - \alpha_{A_t}^*| \,|\, A_{t+k} = a]\right) \quad \text{and} \quad \sigma_B^2 \leq B^2.$$

Here, the constant $C$ will depend on how much $M(\cdot|a)$ differs from the ideal case in which $\hat{Q}(\cdot)$ is the true quantile function for $S(X_t, Y_t)|A_t = a$. In this case we would have that $M(\cdot|a)$ is the linear function $M(p|a) = p, \forall p \in [0,1]$ and $C \leq 2$.

We remark that the term $\mathbb{E}[|\alpha_{A_{t+1}}^* - \alpha_{A_t}^*| \,|\, A_{t+k} = a]$ can be seen as a quantitative measurement of the size of the distribution shift in terms of the change in the critical value $\alpha_a^*$. Thus, we interpret these results as showing that if the distribution shift is small and $\forall a \in \mathcal{A}$, $\hat{Q}(\cdot)$ gives reasonable coverage of the distribution of $S(X_t, Y_t)|A_t = a$, then $T^{-1}\sum_{t=1}^T \text{err}_t$ will concentrate well around $\alpha$.

### 4.2.3 Achieving approximate marginal coverage

Theorem 4.1 bounds the distance between the average miscoverage rate and the target level over long stretches of time. On the other hand, it provides no information about the marginal coverage frequency at a single time step. The following result shows that if the distribution shift is small, the realized marginal coverage rate $M(\alpha_t|A_t)$ will be close to $\alpha$ on average.

**Theorem 4.2** *Assume that there exists a constant $L > 0$ such that for all $a \in \mathcal{A}$ and all $\alpha_1, \alpha_2 \in \mathbb{R}$,*

$$|M(\alpha_2|a) - M(\alpha_1|a)| \leq L|\alpha_2 - \alpha_1|.$$

*Assume additionally that for all $a \in \mathcal{A}$ there exists $\alpha_a^* \in (0,1)$ such that $M(\alpha_a^*|a) = \alpha$. Then,*

$$\mathbb{E}[(M(\alpha_t|A_t) - \alpha)^2] \leq \frac{L(1+\gamma)}{\gamma}\mathbb{E}[|\alpha_{A_{t+1}}^* - \alpha_{A_t}^*|] + \frac{L}{2}\gamma. \tag{7}$$

Once again we emphasize that (7) holds for any choice of $t$ owing to the fact that $(\alpha_t, A_t, \text{err}_t)$ is assumed stationary and thus the quantities appearing in the bound are invariant across $t$. Proof of this result can be found in Section A.8 of the Appendix. We remark that the right-hand side of (7) is minimized by choosing $\gamma = (2\mathbb{E}[|\alpha_{A_{t+1}}^* - \alpha_{A_t}^*|])^{1/2}$, which gives the inequality

$$\mathbb{E}[(M(\alpha_t|A_t) - \alpha)^2] \leq L(\sqrt{2} + 1)\sqrt{\mathbb{E}[|\alpha_{A_{t+1}}^* - \alpha_{A_t}^*|]}.$$

As above we have that in the ideal case $\hat{Q}(\cdot)$ is a perfect estimate of the quantiles of $S(X_t, Y_t)|A_t = a$ and thus $M(p|a) = p$ and $L = 1$. Moreover, we once again have the interpretation that $\mathbb{E}[|\alpha_{A_{t+1}}^* - \alpha_{A_t}^*|]$ is a quantitative measurement of the distribution shift. Thus, this result can be interpreted as bounding the average difference between the realized and target marginal coverage in terms of the size of the underlying distribution shift. Finally, note that the choice $\gamma = (2\mathbb{E}[|\alpha_{A_{t+1}}^* - \alpha_{A_t}^*|])^{1/2}$ formalizes our intuition that $\gamma$ should be chosen to be larger in domains with greater distribution shift, while not being so large as to cause $\alpha_t$ to be overly volatile.

## 5  Impact of $S_t(\cdot)$ on the performance

The performance of all conformal inference methods depends heavily on the design of the conformity score. Previous work has shown how carefully chosen scores or even explicit optimization of the

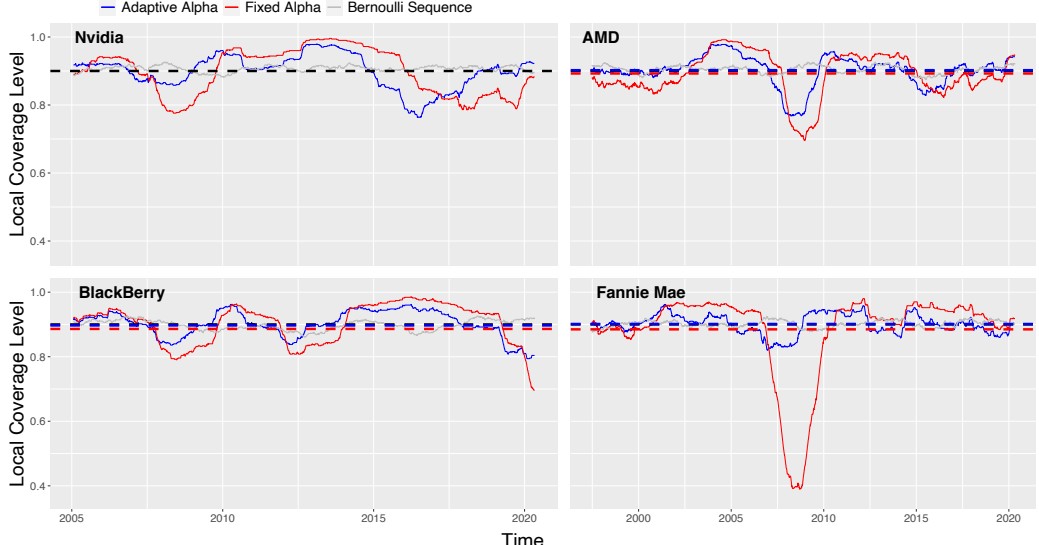

Figure 2: Local coverage frequencies for adaptive conformal (blue), a non-adaptive method that holds $\alpha_t = \alpha$ fixed (red), and an i.i.d. Bernoulli(0.1) sequence (grey) for the prediction of stock market volatility with conformity score $\tilde{S}_t$. The coloured dotted lines mark the average coverage obtained across all time points, while the black line indicates the target level of $1 - \alpha = 0.9$.

interval width can be used to obtain smaller prediction sets [e.g. 28, 29, 20, 12]. Adaptive conformal inference can work with any conformity score $S_t(\cdot)$ and quantile function $\hat{Q}_t(\cdot)$ and thus can be directly combined with other improvements in conformal inference to obtain shorter intervals. One important caveat here is that the lengths of conformal prediction sets depend directly on the quality of the fitted regression model. Thus, to obtain smaller intervals one should re-fit the model at each time step using the most recent data to build the most accurate predictions. This is exactly what we have done in our experiments in Sections 2.2 and 6.

In addition to this, the choice of $S_t(\cdot)$ can also have a direct effect on the coverage properties of adaptive conformal inference. Theorems 4.1 and 4.2 show that the performance of adaptive conformal inference is controlled by the size of the shift in the optimal parameter $\alpha_t^*$ across time. Moreover, $\alpha_t^*$ itself is in one-to-one correspondence with the $1 - \alpha$ quantile of $S_t(X_t, Y_t)$. Thus, the coverage properties of adaptive conformal inference depend on how close $S_t(X_t, Y_t)$ is to being stationary.

For a simple example illustrating the impact of this dependence, note that in Section 2.2 we formed prediction sets using the conformity score

$$S_t := \frac{|V_t - \hat{\sigma}_t^2|}{\hat{\sigma}_t^2}.$$

An *a priori* reasonable alternative to this is the unnormalized score

$$\tilde{S}_t := |V_t - \hat{\sigma}_t^2|.$$

However, after a more careful examination it becomes unsurprising that normalization by $\hat{\sigma}_t^2$ is critical for obtaining an approximately stationary conformity score and thus $\tilde{S}_t$ leads to much worse coverage properties. Figure 2 shows the local coverage frequency (see (4)) of adaptive conformal inference using $\tilde{S}_t$. In comparison to Figure 1 the coverage now undergoes much wider swings away from the target level of 0.9. This issue can be partially mitigated by choosing a larger value of $\gamma$ that gives greater adaptivity to the algorithm.

## 6 Real data example: election night predictions

During the 2020 US presidential election *The Washington Post* used conformalized quantile regression (CQR) (see (1) and Section 1.1) to produce county level predictions of the vote total on election night [13]. Here we replicate the core elements of this method using both fixed and adaptive quantiles.

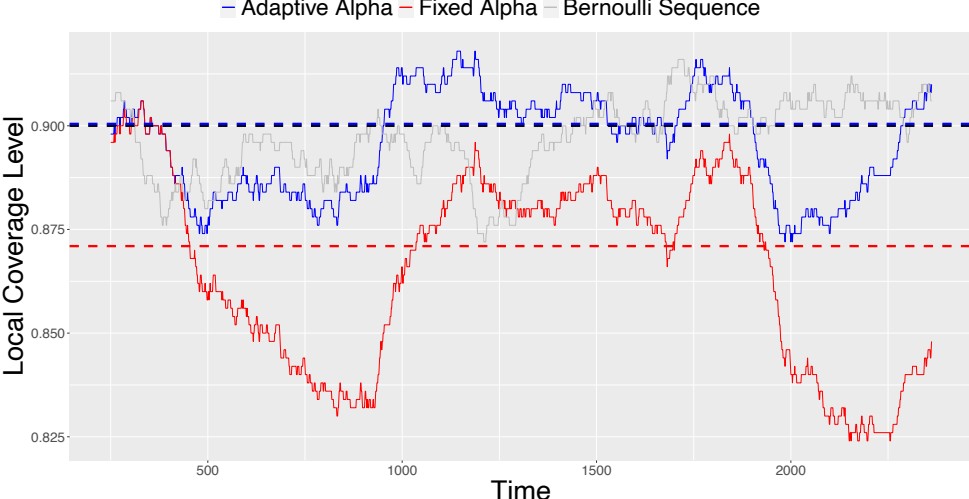

Figure 3: Local coverage frequencies of adaptive conformal (blue), a non-adaptive method that holds $\alpha_t = \alpha$ fixed (red), and an i.i.d. Bernoulli(0.1) sequence (grey) for county-level election predictions. Coloured dotted lines show the average coverage across all time points, while the black line indicates the target coverage level of $1 - \alpha = 0.9$.

To make the setting precise, let $\{Y_t\}_{1 \leq t \leq T}$ denote the number of votes cast for presidential candidate Joe Biden in the 2020 election in each of approximately $T = 3000$ counties in the United States. Let $X_t$ denote a set of demographic covariates associated to the $t$th county. In our experiment $X_t$ will include information on the make-up of the county population by ethnicity, age, sex, median income and education (see Section A.6.1 for details). On election night county vote totals were observed as soon as the vote count was completed. If the order in which vote counts completed was uniformly random $\{(X_t, Y_t)\}_{1 \leq t \leq T}$ would be an exchangeable sequence on which we could run standard conformal inference methods. In reality, larger urban counties tend to report results later than smaller rural counties and counties on the east coast of the US report earlier than those on the west coast. Thus, the distribution of $(X_t, Y_t)$ can be viewed as drifting throughout election night.

We apply CQR to predict the county-level vote totals (see Section A.6.2 for details). To replicate the east to west coast bias observed on election night we order the counties by their time zone with eastern time counties appearing first and Hawaiian counties appearing last. Within each time zone counties are ordered uniformly at random. Figure 3 shows the realized local coverage frequency over the most recent 300 counties (see (4)) for the non-adaptive and adaptive conformal methods. We find that the non-adaptive method fails to maintain the desired 90% coverage level, incurring large troughs in its coverage frequency during time zone changes. On the other hand, the adaptive method maintains approximate 90% coverage across all time points with deviations in its local coverage level comparable to what is observed in Bernoulli sequences.

## 7 Discussion

There are still many open problems in this area. The methods we develop are specific to cases where $Y_t$ is revealed at each time point. However, there are many settings in which we receive the response in a delayed fashion or in large batches. In addition, our theoretical results in Section 4.2 are limited to a single model for the data generating distribution and the special case where the quantile function $\hat{Q}_t(\cdot)$ is fixed across time. It would be interesting to determine if similar results can be obtained in settings where $\hat{Q}_t(\cdot)$ is fit in an online fashion on the most recent data. Another potential area for improvement is in the choice of the step size $\gamma$. In Section 2.1 we give some heuristic guidelines for choosing $\gamma$ based on the size of the distribution shift in the environment. Ideally however we would like to be able to determine $\gamma$ adaptively without prior knowledge. Finally, our experimental results are limited to just two domains. Additional work is needed to determine if our methods can successfully protect against a wider variety of real-world distribution shifts.

# 8 Acknowledgements

E.C. was supported by Office of Naval Research grant N00014-20-12157, by the National Science Foundation grants OAC 1934578 and DMS 2032014, by the Army Research Office (ARO) under grant W911NF-17-1-0304, and by the Simons Foundation under award 814641. We thank John Cherian for valuable discussions related to Presidential Election Night 2020 and Lihua Lei for helpful comments on the connection to online learning.

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
