# A Appendix

## A.1 Connection to online learning

In Section 2 we motivated the update (2) as a way to adjust the size of our prediction sets in response to the realized historical miscoverage frequency. Alternatively, one could also derive (2) as an online gradient descent algorithm with respect to the pinball loss. To be more precise let

$$\beta_t := \sup\{\beta : Y_t \in \hat{C}_t(\beta)\},$$

where we remark that $\hat{C}_t(\beta_t)$ can be thought of as the smallest prediction set containing $Y_t$. Additionally, define the pinball loss

$$\rho_\alpha(u) = \begin{cases} \alpha u, \ u > 0, \\ -(1-\alpha)u, \ u \leq 0. \end{cases}.$$

and consider the loss $\ell(\alpha_t, \beta_t) = \rho_\alpha(\beta_t - \alpha_t)$. Then, one directly computes that

$$\alpha_t - \gamma \partial_{\alpha_t} \ell(\alpha_t, \beta_t) = \alpha_t + \gamma(\alpha - \mathbb{1}_{\alpha_t > \beta_t}) = \alpha_t + \gamma(\alpha - \mathrm{err}_t).$$

Because the pinball loss is convex, this gradient descent update falls within a well understood class of algorithms that have been extensively studied in the online learning literature (see e.g. [17]). A standard analysis may then be to bound the regret of $\alpha_t$ defined as

$$\mathrm{Reg}_T := \sum_{t=1}^{T} \ell(\alpha_t, \beta_t) - \min_\beta \sum_{t=1}^{T} \ell(\beta, \beta_t).$$

Unfortunately, this notion of regret fails to capture our intuition that $\alpha_t$ is adaptively tracking the moving target $\alpha_t^*$. Thus, we make the connection to online gradient descent only in passing and develop alternative theoretical tools in Section 4.

## A.2 Stock prices

Figure 4 shows daily open prices for the four stocks considered in Section 2.2.

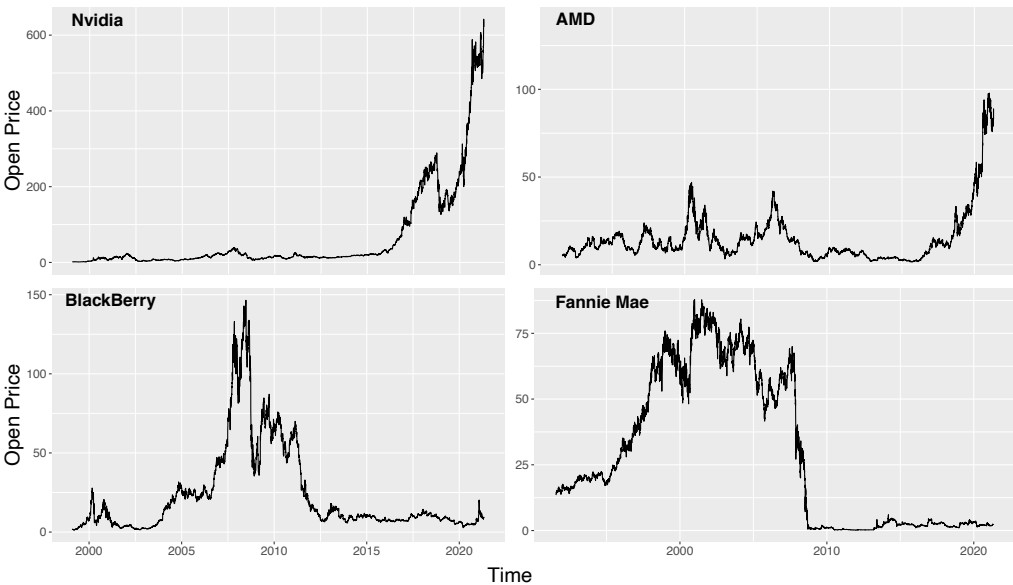

Figure 4: Daily open prices for the four stocks considered in Section 2.2.

## A.3 Trajectories of $\alpha_t$

In this section we show the realized trajectories of $\alpha_t$ obtained in our experiments from Sections 2.2 and 6.

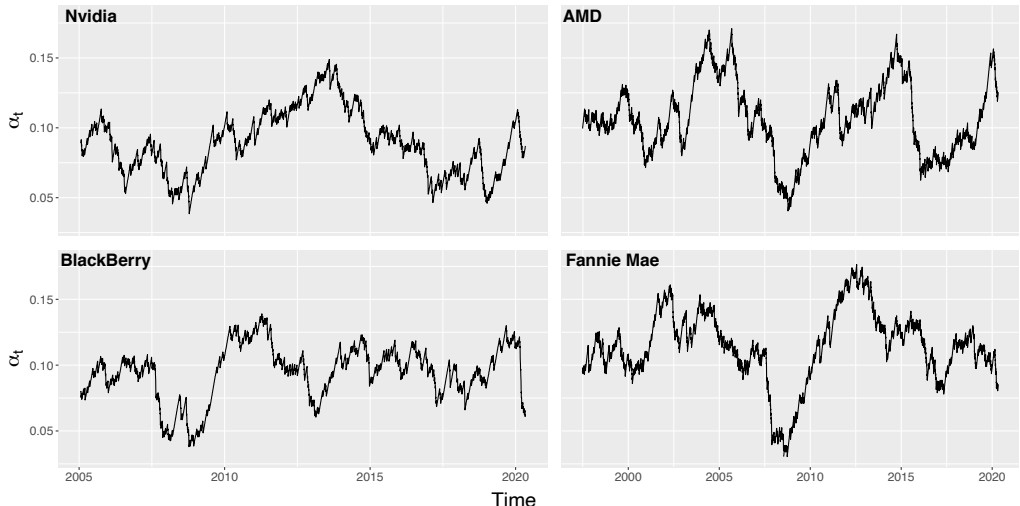

Figure 5: Realized trajectories of $\alpha_t$ for predicting stock market volatility as outlined in Section 2.2 using update (2).

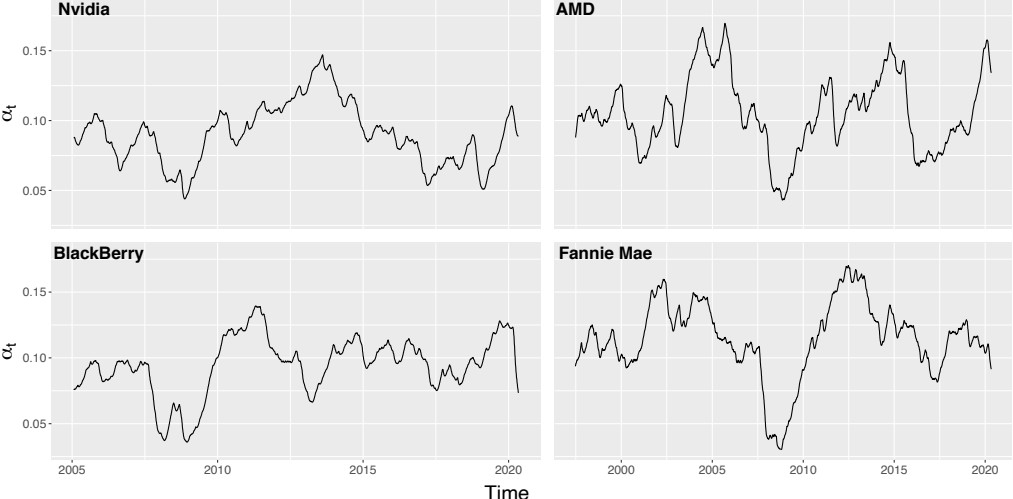

Figure 6: Realized trajectories of $\alpha_t$ for predicting stock market volatility as outlined in Section 2.2 using update (3) with $w_s \propto 0.95^{t-s}$.

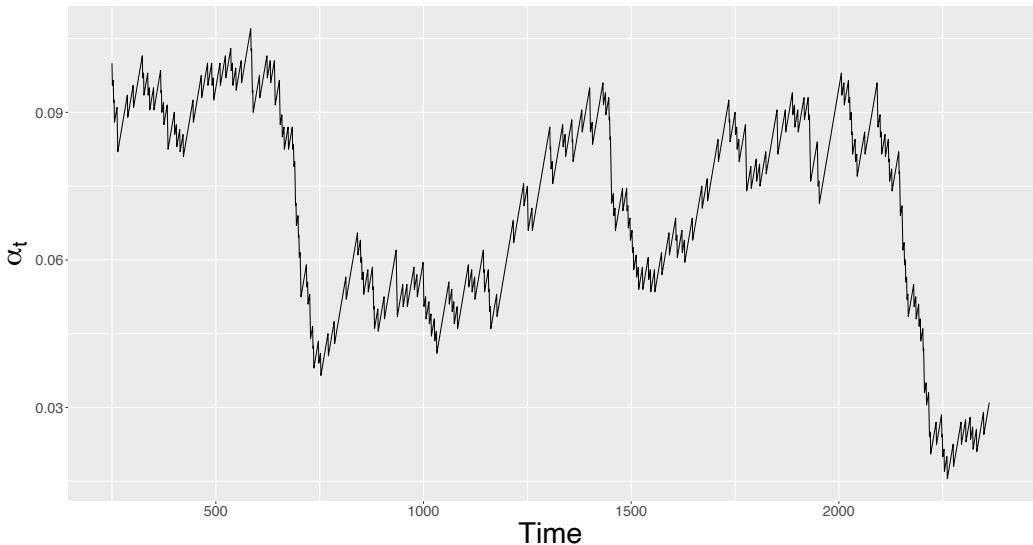

Figure 7: Realized trajectory of $\alpha_t$ for election night forecasting as outlined in Section 6 using update (2).

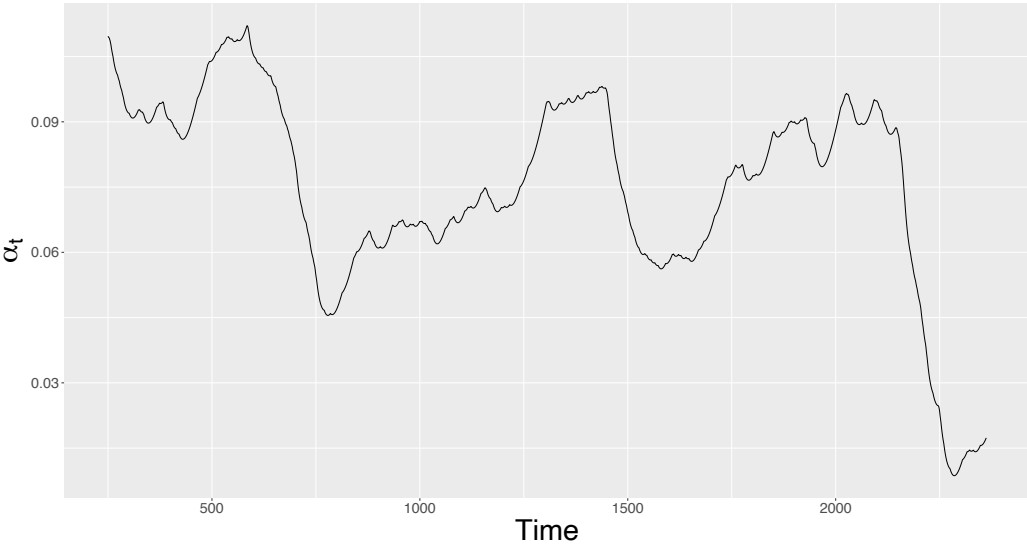

Figure 8: Realized trajectory of $\alpha_t$ for election night forecasting as outlined in Section 6 using update (3) with $w_s \propto 0.95^{t-s}$.

### A.4 Coverage for additional stocks

Figure 9 shows the local coverage level of adaptive and non-adaptive conformal inference for the prediction of market volatility (see Section 2.2) for 8 additional stocks/indices.

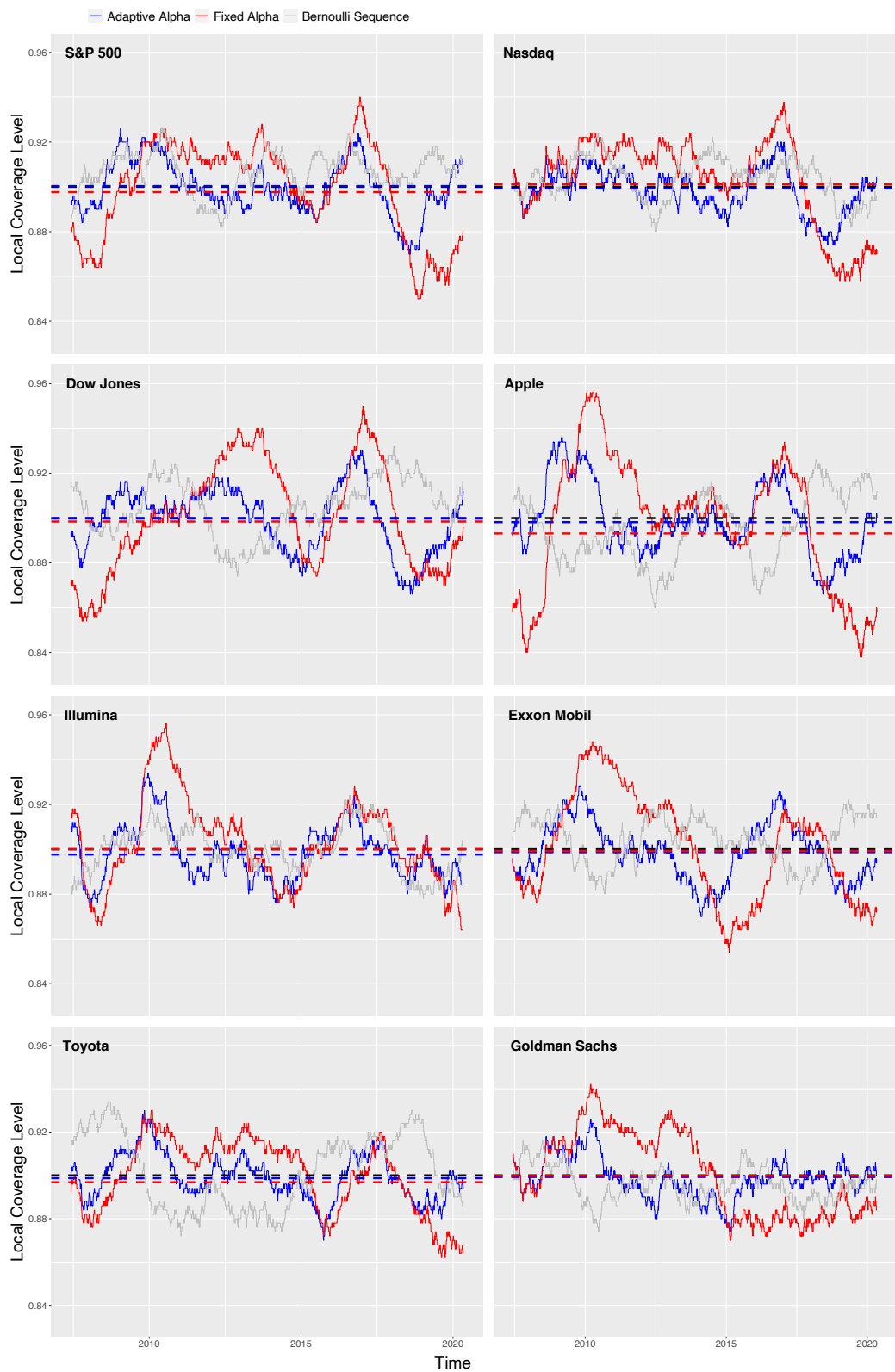

Figure 9: Local coverage frequencies for adaptive conformal (blue), a non-adaptive method that holds $\alpha_t = \alpha$ fixed (red), and an i.i.d. Bernoulli(0.1) sequence (grey) for the prediction of market volatility. The coloured dotted lines mark the average coverage obtained across all time points, while the black line indicates the target level of $1 - \alpha = 0.9$.

## A.5  Existence of a stationary distribution for $(\alpha_t, A_t)$

In this section we give one simple example in which the Markov chain $\{(\alpha_t, A_t)\}_{t\in\mathbb{N}}$ will have a unique stationary distribution. The setting considered here is the same as the one described in Section 4.2.

Let $\alpha_t$ be initialized with $\alpha_1 \in \{\alpha + k\gamma\alpha : k \in \mathbb{Z}\}$. Assume that $\mathcal{A}$ is finite. Let $P$ be the transition matrix of $\{A_t\}_{t\in\mathbb{N}}$ and assume that for all $a_1, a_2 \in \mathcal{A}$, $P_{a_1,a_2} = \mathbb{P}(A_{t+1} = a_2 | A_t = a_1) > 0$. Assume that $\alpha$ satisfies $\alpha^{-1}(1-\alpha) \in \mathbb{N}$. This will hold for common choices of $\alpha$ such as $\alpha = 0.1$ and $\alpha = 0.05$. Finally, assume that for all $a \in \mathcal{A}$ and all $p \in (0,1)$, $\mathbb{P}(S(X_t, Y_t) \leq \hat{Q}(p) | A = a) \in (0,1)$. This will occur for example when $S(X_t, Y_t) | A_t = a$ is supported on $\mathbb{R}$ and $\hat{Q}(\cdot)$ is finite valued for all $p \in (0,1)$.

We claim that in this case $\{(\alpha_t, A_t)\}_{t\in\mathbb{N}}$ has a unique stationary distribution. To prove this it is sufficient to show that this chain is irreducible and has a finite state space. To check that it has a finite state space it is sufficient to show that $\alpha_t$ has a finite state space. We claim that with probability one we have that for all $t \in \mathbb{N}$, $\alpha_t \in \{\alpha + k\gamma\alpha : k \in \mathbb{Z}\}$. To prove this we proceed by induction. The base case is given by our choice of $\alpha_1$. For the inductive step note that

$$\alpha_t \in \{\alpha + k\gamma\alpha : k \in \mathbb{Z}\}$$
$$\implies \alpha_{t+1} = \alpha_t + \gamma(\alpha - \text{err}_t) = \begin{cases} \alpha_t + \gamma\alpha, & \text{if err}_t = 0, \\ \alpha_t - \gamma\alpha(\alpha^{-1}(1-\alpha)), & \text{if err}_t = 1, \end{cases} \in \{\alpha + k\gamma\alpha : k \in \mathbb{Z}\}.$$

Since $\alpha_t$ is also bounded (see Lemma 4.1) this implies that $\alpha_t$ has a finite state space.

Finally, the fact that $\{(\alpha_t, A_t)\}_{t\in\mathbb{N}}$ is irreducible follows easily from our assumptions on $\{A_t\}_{t\in\mathbb{N}}$, $\hat{Q}(\cdot)$, and $\mathbb{P}(S(X_t, Y_t) \leq \hat{Q}(p) | A = a)$.

## A.6  Additional information for Section 6

### A.6.1  Dataset description

The county-level demographic characteristics used for prediction were the proportion of the total population that fell into each of the following race categories (either alone or in combination): black or African American, American Indian or Alaska Native, Asian, Native Hawaiian or other Pacific islander. In addition to this, we also used the proportion of the total population that was male, of Hispanic origin, and that fell within each of the age ranges 20-29, 30-44, 45-64, and 65+. Demographic information was obtained from 2019 estimates published by the United States Census Bureau and available at [5]. To supplement these demographic features we also used the median household income and the percentage of individuals with a bachelors degree or higher as covariates. Data on county-level median household incomes was based on 2019 estimates obtained from [7]. The percentage of individuals with a bachelors degree or higher was computed based on data collected during the years 2015-2019 and published at [6]. As an aside, we remark that we used 2019 estimates because this was the most recent year for which data was available.

Vote counts for the 2016 election were obtained from [14], while 2020 election data was taken from [25]. In total, matching covariate and election vote count data were obtained for 3111 counties.

### A.6.2  Detailed prediction algorithm

Algorithm 1 below outlines the core conformal inference method used to predict election results. An R implementation of this algorithm as well as the core method outlined in Section 2.2 can be found at `https://github.com/isgibbs/AdaptiveConformal`.

## A.7  Large deviation bounds for the error sequence

In this section we prove Theorem 4.1. So, throughout we define the sequences $\{\alpha_t\}_{t\in\mathbb{N}}$, $\{\text{err}_t\}_{t\in\mathbb{N}}$, and $\{A_t\}_{t\in\mathbb{N}}$ as in Section 4.2 and we assume that $\{(\alpha_t, A_t)\}_{t\in\mathbb{N}}$ is a stationary Markov chain from which it follows immediately that $\{(\alpha_t, A_t, \text{err}_t)\}_{t\in\mathbb{N}}$ is also stationary. Additionally, we assume that $\hat{Q}(\cdot)$ and $S(\cdot)$ are fixed functions such that $\hat{Q}(\cdot)$ is non-decreasing with $\hat{Q}(x) = -\infty$ for all $x < 0$ and $\hat{Q}(x) = \infty$ for all $x > 1$. The proof of Theorem 4.1 will rely on the following lemmas.

**Algorithm 1:** *CQR method for election night prediction*

---

**Data:** Observed sequence of county-level votes counts and covariates $\{(X_t, Y_t)\}_{1 \leq t \leq T}$ and vote counts for the democratic candidate in the previous election $\{Y_t^{\text{prev}}\}_{1 \leq t \leq T}$.

**for** $t = 1, 2, \ldots, T$ **do**
  Compute the residual $r_t = (Y_t - Y_t^{\text{prev}})/Y_t^{\text{prev}}$

**for** $t = 501, 502, \ldots, T$ **do**
  `// We start making predictions once 500 counties have been observed.`
  Randomly split the data $\{(X_l, r_l)\}_{1 \leq l \leq t-1}$ into a training set $\mathcal{D}_{\text{train}}$ and a calibration set $\mathcal{D}_{\text{cal}}$ with $|\mathcal{D}_{\text{train}}| = \lfloor (t-1) \cdot 0.75 \rfloor$;
  Fit a linear quantile regression model $\hat{q}(x; p)$ on $\mathcal{D}_{\text{train}}$;
  **for** $(X_l, r_l) \in \mathcal{D}_{cal}$ **do**
    Compute the conformity score $S_l = \max\{\hat{q}(X_l; \alpha/2) - r_l, r_l - \hat{q}(X_l; 1 - \alpha/2)\}$;

  Define the quantile function $\hat{Q}_t(p) = \inf \left\{ x : \left( \frac{1}{|\mathcal{D}_{\text{cal}}|} \sum_{(X_l, r_l) \in \mathcal{D}_{\text{cal}}} \mathbb{1}_{S_l \leq x} \right) \geq p \right\}$;
  Return the prediction set
  $\hat{C}_t(\alpha) := \{y : \max\{\hat{q}(X_t; \alpha/2) - \frac{y - Y_t^{\text{prev}}}{Y_t^{\text{prev}}}, \frac{y - Y_t^{\text{prev}}}{Y_t^{\text{prev}}} - \hat{q}(X_t; 1 - \alpha/2)\} \leq \hat{Q}_t(1 - \alpha)\}$;

---

**Lemma A.1** *Let $f : \mathbb{R} \to \mathbb{R}$ and $g : \mathbb{R} \to \mathbb{R}$ be bounded functions such that either*

1. *$f$ is non-increasing and $g$ is non-decreasing,*

2. *or $f$ is non-decreasing and $g$ is non-increasing.*

*Then, for any random variable $Y$*

$$\mathbb{E}[f(Y)g(Y)] \leq \mathbb{E}[f(Y)]\mathbb{E}[g(Y)].$$

The proof of this result is straightforward and can be found in Section A.10.

**Lemma A.2** *For any $\lambda \in \mathbb{R}$ and $t \in \mathbb{N}$ we have that*

$$\mathbb{E}\left[ \prod_{s=1}^{t} \exp(\lambda(\text{err}_s - \mathbb{E}[\text{err}_s | A_s])) \right] \leq \exp(\lambda^2/2) \mathbb{E}\left[ \prod_{s=1}^{t-1} \exp(\lambda(\text{err}_s - \mathbb{E}[\text{err}_s | A_s])) \right]. \quad (8)$$

**Proof:** By conditioning on $\alpha_1$ and $A_1, \ldots, A_t$ on both the left and right-hand side of (8) we may view these quantities as fixed. Thus, while for readability we do not denote this conditioning explicitly, the following calculations should be read as conditional on $\alpha_1$ and $A_1, \ldots, A_t$. Then,

$$\mathbb{E}\left[ \prod_{s=1}^{t} \exp(\lambda(\text{err}_s - \mathbb{E}[\text{err}_s | A_s])) \right]$$
$$= \mathbb{E}\left[ \prod_{s=1}^{t-1} \exp(\lambda(\text{err}_s - \mathbb{E}[\text{err}_s | A_s])) \mathbb{E}\left[ \exp(\lambda(\text{err}_t - \mathbb{E}[\text{err}_t | A_t])) \middle| \text{err}_1, \ldots, \text{err}_{t-1} \right] \right].$$

Recall that $\alpha_t = \alpha_1 + \gamma \sum_{s=1}^{t-1}(\alpha - \text{err}_s)$ is a deterministic function of $\alpha_1$ and $\sum_{s=1}^{t-1} \text{err}_s$. So, we may define the functions $f(\sum_{s=1}^{t-1} \text{err}_s) = \prod_{s=1}^{t-1} \exp(\lambda(\text{err}_s - \mathbb{E}[\text{err}_s | A_s]))$ and

$$g(\sum_{s=1}^{t-1} \text{err}_s) := \mathbb{E}\left[ \exp(\lambda(\text{err}_t - \mathbb{E}[\text{err}_t | A_t])) | \text{err}_1, \ldots, \text{err}_{t-1} \right]$$

$$= P_{A_t}(S(X_t, Y_t) \leq \hat{Q}(1 - \alpha_t)) \exp(-\lambda \mathbb{E}[\text{err}_t | A_t])$$
$$+ (1 - P_{A_t}(S(X_t, Y_t) \leq \hat{Q}(1 - \alpha_t))) \exp(\lambda(1 - \mathbb{E}[\text{err}_t | A_t])),$$

where we emphasize that on the last line $A_t$ and $\alpha_t$ should be viewed as fixed quantities. Now, since $\alpha_t$ is monotonically decreasing in $\sum_{s=1}^{t-1} \text{err}_s$ it should be clear that if $\lambda \geq 0$ then $g$ is non-increasing

(resp. non-decreasing for $\lambda < 0$) and $f$ is non-decreasing (resp. non-increasing for $\lambda < 0$). So, by Lemma A.1 we have that

$$\mathbb{E}\left[\prod_{s=1}^{t-1}\exp(\lambda(\mathrm{err}_s - \mathbb{E}[\mathrm{err}_s|A_s]))\mathbb{E}\left[\exp(\lambda(\mathrm{err}_t - \mathbb{E}[\mathrm{err}_t|A_t]))\Big|\mathrm{err}_1,\ldots,\mathrm{err}_{t-1}\right]\right]$$

$$\leq \mathbb{E}\left[\prod_{s=1}^{t-1}\exp(\lambda(\mathrm{err}_s - \mathbb{E}[\mathrm{err}_s|A_s]))\right]\mathbb{E}\left[\mathbb{E}\left[\exp(\lambda(\mathrm{err}_t - \mathbb{E}[\mathrm{err}_t|A_t]))\Big|\mathrm{err}_1,\ldots,\mathrm{err}_{t-1}\right]\right]$$

$$= \mathbb{E}\left[\prod_{s=1}^{t-1}\exp(\lambda(\mathrm{err}_s - \mathbb{E}[\mathrm{err}_s|A_s]))\right]\mathbb{E}[\exp(\lambda(\mathrm{err}_t - \mathbb{E}[\mathrm{err}_t|A_t]))]$$

$$\leq \mathbb{E}\left[\prod_{s=1}^{t-1}\exp(\lambda(\mathrm{err}_s - \mathbb{E}[\mathrm{err}_s|A_s]))\right]\exp(\lambda^2/2),$$

where the last inequality follows by Hoeffding's lemma (see Lemma A.5). $\square$

The final result we will need in order to prove Theorem 4.1 is a large deviation bound for Markov chains.

**Definition A.1** *Let $\{X_i\}_{i\in\mathbb{N}} \subseteq \mathcal{X}$ be a Markov chain with transition kernel $P$ and stationary distribution $\pi$. Define the inner product space*

$$L^2(\mathcal{X},\pi) = \left\{h : \int_{\mathcal{X}} h(x)^2\pi(dx) < \infty\right\}$$

*with inner product*

$$\langle h_1, h_2\rangle_\pi = \int_{\mathcal{X}} h_1(x)h_2(x)\pi(dx).$$

*For any $h \in L^2(\mathcal{X},\pi)$, let*

$$L^2(\mathcal{X},\pi) \ni Ph := \int h(y)P(\cdot,dy).$$

*Then, we say that $\{X_i\}_{i\in\mathbb{N}}$ has non-zero absolute spectral gap $1 - \eta$ if*

$$\eta := \sup\left\{\sqrt{\langle Ph, Ph\rangle_\pi} : \langle h, h\rangle_\pi = 1, \int_{\mathcal{X}} h(x)\pi(dx) = 0\right\} < 1.$$

**Theorem A.1 (Theorem 1 in [19])** *Let $\{X_i\}_{i\in\mathbb{N}} \subseteq \mathcal{X}$ be a stationary Markov chain with invariant distribution $\pi$ and non-zero absolute spectral gap $1 - \eta > 0$. Let $f_i : \mathcal{X} \to [-C, C]$ be a sequence of functions with $\pi(f_i) = 0$ and define $\sigma^2 := \sum_{i=1}^{n} \pi(f_i^2)/n$. Then, for $\forall\epsilon > 0$,*

$$\mathbb{P}_\pi\left(\frac{1}{n}\sum_{i=1}^{n} f_i(X_i) \geq \epsilon\right) \leq \exp\left(-\frac{n(1-\eta)\epsilon^2/2}{(1+\eta)\sigma^2 + 5C\epsilon}\right).$$

Finally, we are ready to prove Theorem 4.1.

**Proof:** [Proof of Theorem 4.1] Write

$$\mathbb{P}\left(\left|\frac{1}{T}\sum_{t=1}^{T}\mathrm{err}_t - \alpha\right| > \epsilon\right) \tag{9}$$

$$\leq \mathbb{P}\left(\left|\frac{1}{T}\sum_{t=1}^{T}\mathrm{err}_t - \mathbb{E}[\mathrm{err}_t|A_t]\right| > \epsilon/2\right) + \mathbb{P}\left(\left|\frac{1}{T}\sum_{t=1}^{T}\mathbb{E}[\mathrm{err}_t|A_t] - \alpha\right| > \epsilon/2\right) \tag{10}$$

By applying lemma A.2 inductively we have that for all $\lambda > 0$,

$$\mathbb{P}\left(\frac{1}{T}\sum_{t=1}^{T}\mathrm{err}_t - \mathbb{E}[\mathrm{err}_t|A_t] > \epsilon/2\right) \leq \exp(-T\lambda\epsilon/2)\mathbb{E}\left[\prod_{t=1}^{T}\exp(\lambda(\mathrm{err}_t - \mathbb{E}[\mathrm{err}_t|A_t]))\right]$$

$$\leq \exp(-T\lambda\epsilon/2)\exp(T\lambda^2/2),$$

with an identical bound on the left tail. Choosing $\lambda = \epsilon/2$ gives the bound

$$\mathbb{P}\left(\left|\frac{1}{T}\sum_{t=1}^{T}\mathrm{err}_t - \mathbb{E}[\mathrm{err}_t|A_t]\right| > \epsilon/2\right) \le 2\exp(-T\epsilon^2/8).$$

On the other hand, the second term in (10) can be bounded directly using Theorem A.1. □

## A.8 Approximate marginal coverage

In this section we prove Theorem 4.2.

**Proof:** [Proof of Theorem 4.2] Our proof follows similar steps to those presented in previous works on stochastic gradient descent under distribution shift [9, 35]. First, note that

$$(\alpha_{t+1} - \alpha^*_{A_t})^2 = (\alpha_t - \alpha^*_{A_t})^2 + 2\gamma(\alpha - \mathrm{err}_t)(\alpha_t - \alpha^*_{A_t}) + \gamma^2(\alpha - \mathrm{err}_t)^2.$$

Now recalling that $M(\alpha^*_{A_t}|A_t) = \alpha$, we find that

$$
\begin{aligned}
-\mathbb{E}[(\alpha - \mathrm{err}_t)(\alpha_t - \alpha^*_{A_t})] &= \mathbb{E}[\mathbb{E}[(\mathrm{err}_t - \alpha)(\alpha_t - \alpha^*_{A_t})|A_t, \alpha_t]] \\
&= \mathbb{E}[(M(\alpha_t|A_t) - M(\alpha^*_{A_t}|A_t))(\alpha_t - \alpha^*_{A_t})] \\
&\ge \frac{1}{L}\mathbb{E}[(M(\alpha_t|A_t) - M(\alpha^*_{A_t}|A_t))^2] \\
&= \frac{1}{L}\mathbb{E}[(M(\alpha_t|A_t) - \alpha)^2].
\end{aligned}
$$

Thus it follows that

$$
\begin{aligned}
2\gamma L^{-1}\sum_{t=1}^{T}\mathbb{E}[(M(\alpha_t|A_t) - \alpha)^2] &\le \sum_{t=1}^{T}\mathbb{E}[(\alpha_t - \alpha^*_{A_t})^2 - (\alpha_{t+1} - \alpha^*_{A_t})^2 + \gamma^2(\alpha - \mathrm{err}_t)^2] \\
&\le \sum_{t=1}^{T}\mathbb{E}[(\alpha_{t+1} - \alpha^*_{A_{t+1}})^2 - (\alpha_{t+1} - \alpha^*_{A_t})^2] + \mathbb{E}[(\alpha_1 - \alpha^*_{A_1})^2] + \gamma^2 T \\
&\le \sum_{t=1}^{T}\mathbb{E}[2\alpha_{t+1}(\alpha^*_{A_t} - \alpha^*_{A_{t+1}})] + (\alpha^*_{A_{T+1}})^2 + \mathbb{E}[(\alpha_1 - \alpha^*_{A_1})^2] + \gamma^2 T \\
&\le \sum_{t=1}^{T}2(1+\gamma)\mathbb{E}[|\alpha^*_{A_{t+1}} - \alpha^*_{A_t}|] + (\alpha^*_{A_{T+1}})^2 + \mathbb{E}[(\alpha_1 - \alpha^*_{A_1})^2] + \gamma^2 T,
\end{aligned}
$$

where the last inequality follows from Lemma 4.1. So, re-arranging we get the inequality

$$
\begin{aligned}
&\frac{1}{T}\sum_{t=1}^{T}\mathbb{E}[(M(\alpha_t|A_t) - \alpha)^2] \\
&\le \frac{L}{2T\gamma}\left(\sum_{t=1}^{T}2(1+\gamma)\mathbb{E}[|\alpha^*_{A_{t+1}} - \alpha^*_{A_t}|] + (\alpha^*_{A_{T+1}})^2 + \mathbb{E}[(\alpha_1 - \alpha^*_{A_1})^2] + \gamma^2 T\right).
\end{aligned}
$$

Finally, since $\{(A_t, \alpha_t)\}_{t\in\mathbb{N}}$ is stationary we may let $T \to \infty$ to get that

$$\mathbb{E}[(M(\alpha_t|A_t) - \alpha)^2] \le \frac{L(1+\gamma)}{\gamma}\mathbb{E}[|\alpha^*_{A_{t+1}} - \alpha^*_{A_t}|] + \frac{L\gamma}{2}$$

as claimed. □

## A.9 Bounds on $B$ and $\sigma_B^2$

In this section we bound the constants $B$ and $\sigma_B^2$ appearing in the statement of Theorem 4.1. Let

$$\epsilon_1 = \sup_{k\in\{0,1,2,\dots\}}\sup_{a\in\mathcal{A}}\mathbb{E}[|\alpha^*_{A_{T-k}} - \alpha^*_{A_{T-k-1}}||A_T = a]$$

$$\text{and } \epsilon_2 = \sup_{k\in\{0,1,2,\dots\}}\sup_{a\in\mathcal{A}}\mathbb{E}[(\alpha^*_{A_{T-k}} - \alpha^*_{A_{T-k-1}})^2|A_T = a].$$

Then, our main result is Lemma A.4 which shows that

$$B \leq C(\gamma + \gamma^{-1}(\epsilon_1 + \epsilon_2)) \quad \text{and} \quad \sigma_B^2 \leq B^2, \tag{11}$$

where the constant $C$ depends on how close $M(\cdot|a)$ is to the ideal linear function $M(p|a) = p$.

Plugging (11) into Theorem 4.1 gives a concentration inequality for $|T^{-1}\sum_{t=1}^T \text{err}_t - \alpha|$. In particular, suppose we use an optimal stepsize of $\gamma \propto \sqrt{\epsilon_1}$. Then, combining (11) with Theorem 4.1 roughly tells us that

$$\left| \frac{1}{T} \sum_{t=1}^T \text{err}_t - \alpha \right| \leq O\left( \max\left\{ \frac{1}{\sqrt{T}}, \frac{\sqrt{\epsilon_1}}{\sqrt{T(1-\eta)}}, \frac{\sqrt{\epsilon_1}}{T(1-\eta)} \right\} \right). \tag{12}$$

As a comparison it may be instructive to note that the more naive bound given in Proposition 4.1 can be written as

$$\left| \frac{1}{T} \sum_{t=1}^T \text{err}_t - \alpha \right| \leq O\left( \frac{1}{T\gamma} \right) = O\left( \frac{1}{T\sqrt{\epsilon_1}} \right). \tag{13}$$

A sharp reader may notice that the naive bound given in (13) actually goes to 0 faster in $T$ than the HMM-based bound shown in (12). While this is true, the bound (13) has the highly undesirable property of increasing in $1/\sqrt{\epsilon_1}$, i.e. the bound increases as the size of the distribution shift decreases. On the other hand, the HMM-based bound has the more intuitive property of decreasing with the distribution shift.

The only remaining issue is to determine the size of $\sqrt{\epsilon_1/(1-\eta)}$. To provide some insight into this quantity note that there are two main regimes in which we expect $|T^{-1}\sum_{t=1}^T \text{err}_t - \alpha|$ to be small:

1. Environments in which the state $A_t$ changes frequently, but $|\alpha_{A_{t+1}}^* - \alpha_{A_t}^*|$ is always small. In this case it is reasonable to expect $1 - \eta$ to not be too small and so we anticipate that (12) will give a reasonable bound.

2. Environments in which the state changes very infrequently. In this case $\{A_t\}_{t\in\mathbb{N}}$ will mix slowly and so we expect $1 - \eta$ to be quite small. Additionally, we also have that $\alpha_{A_{t+1}}^* = \alpha_{A_t}^*$ a large proportion of the time and thus $\epsilon_1$ will also be small. As a result, it is not immediately clear what (12) tells us about $|T^{-1}\sum_{t=1}^T \text{err}_t - \alpha|$. Below we give one simple example that demonstrates that (12) can also be a reasonable bound in this instance.

**Example A.1** *Let $\{A_t\}_{t\in\mathbb{N}}$ be the Markov chain with states $\{1, \ldots, n\}$ and transition matrix*

$$P = \left( p - \frac{1-p}{n-1} \right) I + \frac{1-p}{n-1} 11^T,$$

*where $p \in [0,1]$ is taken to be very close to 1. Let $\Delta := \max_{i \neq j} |\alpha_i^* - \alpha_j^*|$. Then, we have that $\epsilon_1, \epsilon_2 \leq \Delta(1-p)$. Moreover, note that this chain has spectral gap $1 - \eta \cong 1 - p$. Thus, (12) simplifies to*

$$\left| \frac{1}{T} \sum_{t=1}^T err_t - \alpha \right| \leq O\left( \max\left\{ \frac{1}{\sqrt{T}}, \frac{\sqrt{\Delta}}{T\sqrt{1-p}} \right\} \right). \tag{14}$$

*In particular, for $T > \Delta/(1-p)$ we find that the error sequence concentrates at a rate of $O(1/\sqrt{T})$, which is consistent with the behaviour of an i.i.d. Bernoulli sequence. Finally, to understand this restriction on $T$ note that given a starting state $j \in \{1, \ldots, n\}$ we expect $\alpha_t$ to contract towards $\alpha_j^*$ at a rate of $(1-\gamma)$ and therefore to have that*

$$\frac{1}{T} \sum_{t=1}^T |\alpha_t - \alpha_j^*| \propto \frac{1}{T} \sum_{t=1}^T (1-\gamma)^{t-1} |\alpha_1 - \alpha_j^*| \leq \frac{\Delta}{T\gamma} = \frac{\sqrt{\Delta}}{T\sqrt{1-p}},$$

*where here we assumed that $\alpha_1 \in \{\alpha_1^*, \ldots, \alpha_n^*\}$. Thus, the second term in the maximum in (14) can be seen as accounting for the rate of convergence of $\alpha_t$ to $\alpha_j^*$ during the time that the Markov chain is in state $j$ and given that $\alpha_1$ starts at some $\alpha_i^*$, $1 \leq i \leq n$.*

We now derive (11).

**Lemma A.3** *Assume that $\exists\, 0 < c < 1/(2\gamma)$ such that for all $a \in \mathcal{A}$ and all $p \in [-\gamma, 1 + \gamma]$,*

$$|M(p|a) - M(\alpha_a^*|a)| \geq c|p - \alpha_a^*|.$$

*Then, for all $a \in \mathcal{A}$, $k \in \{0, 1, 2, \dots\}$, and $T \in \mathbb{N}$*

$$\mathbb{E}[(\alpha_1 - \alpha_{A_1}^*)^2 | A_{1+k} = a]$$
$$\leq (1 - 2c\gamma)^{T-1}\mathbb{E}[(\alpha_1 - \alpha_{A_1}^*)^2 | A_{T+k} = a]$$
$$+ \sum_{t=2}^{T}(1 - 2c\gamma)^{T-t}\left(\gamma^2 + 2(1+\gamma)\mathbb{E}[|\alpha_{A_{t-1}}^* - \alpha_{A_t}^*| | A_{T+k} = a] + \mathbb{E}[(\alpha_{A_{t-1}}^* - \alpha_{A_t}^*)^2 | A_{T+k} = a]\right).$$

*Furthermore, if we assume that $\forall a \in \mathcal{A}$ and $k \in \{0, 1, 2, \dots\}$,*

$$\mathbb{E}[|\alpha_{A_{t-k}}^* - \alpha_{A_{t-k-1}}^*| | A_t = a] \leq \epsilon_1 \text{ and } \mathbb{E}[(\alpha_{A_{t-k}}^* - \alpha_{A_{t-k-1}}^*)^2 | A_t = a] \leq \epsilon_2,$$

*then we find that*

$$\mathbb{E}[(\alpha_1 - \alpha_{A_1}^*)^2 | A_{1+k} = a] \leq \frac{1}{2c\gamma}\left(\gamma^2 + 2(1+\gamma)\epsilon_1 + \epsilon_2\right).$$

**Proof:**  Fix any $T \in \mathbb{N}$. Since $\{(\alpha_t, A_t)\}_{t \in \mathbb{N}}$ is stationary we have that

$$\mathbb{E}[(\alpha_1 - \alpha_{A_1}^*)^2 | A_{1+k}] = \mathbb{E}[(\alpha_T - \alpha_{A_T}^*)^2 | A_{T+k}].$$

Now note that

$$\mathbb{E}[(\alpha_T - \alpha_{A_T}^*)^2 | A_{T+k}] = \mathbb{E}[(\alpha_T - \alpha_{A_{T-1}}^*)^2 | A_{T+k}] + 2\mathbb{E}[(\alpha_T - \alpha_{A_{T-1}}^*)(\alpha_{A_{T-1}}^* - \alpha_{A_T}^*) | A_{T+k}]$$
$$+ \mathbb{E}[(\alpha_{A_{T-1}}^* - \alpha_{A_T}^*)^2 | A_{T+k}]$$
$$\leq \mathbb{E}[(\alpha_T - \alpha_{A_{T-1}}^*)^2 | A_{T+k}] + 2(1+\gamma)\mathbb{E}[|\alpha_{A_{T-1}}^* - \alpha_{A_T}^*| | A_{T+k}]$$
$$+ \mathbb{E}[(\alpha_{A_{T-1}}^* - \alpha_{A_T}^*)^2 | A_{T+k}],$$

where on the last line we have applied Lemma 4.1. The first term above can be bounded as

$$\mathbb{E}[(\alpha_T - \alpha_{A_{T-1}}^*)^2 | A_{T+k}]$$
$$\leq \mathbb{E}[(\alpha_{T-1} - \alpha_{A_{T-1}}^*)^2 | A_{T+k}] + 2\mathbb{E}[\gamma(\alpha - \mathrm{err}_{T-1})(\alpha_{T-1} - \alpha_{A_{T-1}}^*) | A_{T+k}] + \gamma^2,$$

where we additionally have that

$$\mathbb{E}[(\alpha - \mathrm{err}_T)(\alpha_{T-1} - \alpha_{A_{T-1}}^*) | A_{T+k}]$$
$$= \mathbb{E}[(\alpha - \mathbb{E}[\mathrm{err}_{T-1} | A_{T-1}, \alpha_{T-1}, A_{T+k}])(\alpha_{T-1} - \alpha_{A_{T-1}}^*) | A_{T+k}]$$
$$= \mathbb{E}[(M(\alpha_{A_{T-1}}^* | A_{T_1}) - M(\alpha_{T-1} | A_{T-1}))(\alpha_{T-1} - \alpha_{A_{T-1}}^*) | A_{T+k}]$$
$$\leq -c\mathbb{E}[(\alpha_{T-1} - \alpha_{A_{T-1}}^*)^2 | A_{T+k}].$$

Whence,

$$\mathbb{E}[(\alpha_T - \alpha_{A_{T-1}}^*)^2 | A_{T+k}] \leq (1 - 2c\gamma)\mathbb{E}[(\alpha_{T-1} - \alpha_{A_{T-1}}^*)^2 | A_{T+k}] + \gamma^2,$$

and plugging this into our first inequality yields

$$\mathbb{E}[(\alpha_T - \alpha_{A_T}^*)^2 | A_{T+k}] \leq (1 - 2c\gamma)\mathbb{E}[(\alpha_{T-1} - \alpha_{A_{T-1}}^*)^2 | A_{T+k}] + \gamma^2$$
$$+ 2(1+\gamma)\mathbb{E}[|\alpha_{A_{T-1}}^* - \alpha_{A_T}^*| | A_{T+k}] + \mathbb{E}[(\alpha_{A_{T-1}}^* - \alpha_{A_T}^*)^2 | A_{T+k}].$$

Repeating this argument inductively gives

$$\mathbb{E}[(\alpha_T - \alpha_{A_T}^*)^2 | A_{T+k} = a]$$
$$\leq (1 - 2c\gamma)^{T-1}\mathbb{E}[(\alpha_1 - \alpha_{A_1}^*)^2 | A_{T+k} = a]$$
$$+ \sum_{t=2}^{T}(1 - 2c\gamma)^{T-t}\left(\gamma^2 + 2(1+\gamma)\mathbb{E}[|\alpha_{A_{t-1}}^* - \alpha_{A_t}^*| | A_{T+k} = a] + \mathbb{E}[(\alpha_{A_{t-1}}^* - \alpha_{A_t}^*)^2 | A_{T+k} = a]\right).$$

The final part of the lemma follows by sending $T \to \infty$. $\qquad\square$

**Lemma A.4** *Assume that for all $a \in \mathcal{A}$ and $p \in [-\gamma, 1+\gamma]$, $M(\cdot|a)$ admits the second order Taylor expansion*

$$M(p|a) - M(\alpha_a^*|a) = C_a^1(p - \alpha_a^*) + C_{p,a}^2(p - \alpha_a^*)^2,$$

*where $0 < c_1 \leq C_a^1 \leq C_1 < 1/\gamma$ and $|C_{p,a}^2| \leq C_2$. Then, for all $T \in \mathbb{N}$ and $a \in \mathcal{A}$,*

$$|\mathbb{E}[\mathrm{err}_1 | A_1 = a] - \alpha| \leq C_1(1 - \gamma c_1)^{T-1}\mathbb{E}[|\alpha_1 - \alpha_{A_1}^*| | A_T = a]$$
$$+ \sum_{t=1}^{T-1} C_1 C_2 \gamma (1 - c_1\gamma)^{T-t-1}\mathbb{E}[(\alpha_t - \alpha_{A_t^*})^2 | A_T = a]$$
$$+ \sum_{t=1}^{T-1} C_1(1 - c_1\gamma)^{T-t-1}\mathbb{E}[|\alpha_{A_{t+1}}^* - \alpha_{A_t}^*| | A_T] + C_2\mathbb{E}[(\alpha_T - \alpha_{A_T}^*)^2 | A_T = a].$$

*Furthermore, suppose the assumptions of Lemma A.3 hold and that $\forall a \in \mathcal{A}$ and $k \in \{0, 1, 2, \dots\}$,*

$$\mathbb{E}[|\alpha_{A_{T-k}}^* - \alpha_{A_{T-k-1}}^*| | A_T = a] \leq \epsilon_1 \text{ and } \mathbb{E}[(\alpha_{A_{T-k}}^* - \alpha_{A_{T-k-1}}^*)^2 | A_T = a] \leq \epsilon_2.$$

*Then $\forall a \in \mathcal{A}$,*

$$|\mathbb{E}[\mathrm{err}_1 | A_1 = a] - \alpha| \leq \left( C_1 C_2 \frac{1}{c_1} + C_2 \right) \frac{1}{2c\gamma}(\gamma^2 + 2(1+\gamma)\epsilon_1 + \epsilon_2) + \frac{C_1}{c_1\gamma}\epsilon_1.$$

**Proof:** Fix any $T \in \mathbb{N}$. Since $\{(\mathrm{err}_t, A_t)\}_{t \in \mathbb{N}}$ is stationary we have that

$$\mathbb{E}[\mathrm{err}_1 | A_1 = a] = \mathbb{E}[\mathrm{err}_T | A_T = a].$$

Then, by Taylor expanding $M(\cdot|A_T)$ we find that

$$|\mathbb{E}[\mathrm{err}_T | A_T = a] - \alpha| = \left|\mathbb{E}[M(\alpha_T | A_T) - M(\alpha_{A_T}^* | A_T) | A_T]\right|$$
$$\leq \left|\mathbb{E}[C_{A_T}^1(\alpha_T - \alpha_{A_T}^*) | A_T]\right| + \mathbb{E}[C_{\alpha_T, A_T}^2(\alpha_T - \alpha_{A_T}^*)^2 | A_T]$$
$$\leq \left|\mathbb{E}[C_{A_T}^1(\alpha_T - \alpha_{A_T}^*) | A_T]\right| + C_2\mathbb{E}[(\alpha_T - \alpha_{A_T}^*)^2 | A_T].$$

The first term above can be further bounded as

$$\left|\mathbb{E}[C_{A_T}^1(\alpha_T - \alpha_{A_T}^*) | A_T]\right|$$
$$= \left|\mathbb{E}[C_{A_T}^1(\alpha_{T-1} + \gamma(\alpha - \mathrm{err}_{T-1}) - \alpha_{A_{T-1}}^*) | A_T] + \mathbb{E}[C_{A_T}^1(\alpha_{A_{T-1}}^* - \alpha_{A_T}^*) | A_T]\right|$$
$$\leq \left|\mathbb{E}[C_{A_T}^1(\alpha_{T-1} - \alpha_{A_{T-1}}^*)] + \gamma\mathbb{E}[C_{A_T}^1(M(\alpha_{A_{T-1}}^* | A_{T-1}) - M(\alpha_{T-1} | A_{T-1})) | A_T]\right|$$
$$\qquad + C_1\mathbb{E}[|\alpha_{A_{T-1}}^* - \alpha_{A_T}^*| | A_T]$$
$$= \left|\mathbb{E}[C_{A_T}^1(1 - \gamma C_{A_{T-1}}^1)(\alpha_{T-1} - \alpha_{A_{T-1}}^*) | A_T]\right| + C_1 C_2 \gamma \mathbb{E}[(\alpha_{T-1} - \alpha_{A_{T-1}}^*)^2 | A_T]$$
$$\qquad + C_1\mathbb{E}[|\alpha_{A_{T-1}}^* - \alpha_{A_T}^*| | A_T].$$

The desired result follows by repeating this process inductively. Finally, the last part of the Lemma follows by sending $T \to \infty$ and applying the result of Lemma A.3.

$\square$

As a final aside we remark that in the main text we claimed that in the ideal case where $M(p|a) = p$ for all $p \in [0, 1]$ this bound can be replaced by

$$|\mathbb{E}[\mathrm{err}_1 | A_1 = a] - \alpha| \leq 2(\gamma + \gamma^{-1}\epsilon_1).$$

This can be justified by using the fact that in this case we have that for all $p \in [-\gamma, 1+\gamma]$

$$M(p|a) - M(\alpha_a^*|a) = (p - \alpha_a^*) + C_{p,a}^2$$

with $|C_{p,a}^2| \leq \gamma$. The desired result then follows by repeating the argument of Lemma A.4.

## A.10  Technical lemmas

**Proof:**  [Proof of Lemma A.1:] We assume without loss of generality that $f$ is non-decreasing and $g$ is non-increasing as otherwise one can simply multiply both $f$ and $g$ by $-1$.

Let $g^U := \sup\{g(y) : f(y) > \mathbb{E}[f(Y)]\}$ and $g^L := \inf\{g(y) : f(y) \leq \mathbb{E}[f(Y)]\}$. By the monotonicity of $f$ and $g$ we clearly have that $g^L \geq g^U$. Therefore,

$$
\begin{aligned}
\mathbb{E}[f(Y)g(Y)] - \mathbb{E}[f(Y)]\mathbb{E}[g(Y)] &= \mathbb{E}[(f(Y) - \mathbb{E}[f(Y)])g(Y)] \\
&\leq \mathbb{E}[(f(Y) - \mathbb{E}[f(Y)])g^L \mathbb{1}_{f(Y) \leq \mathbb{E}[f(Y)]}] + \mathbb{E}[(f(Y) - \mathbb{E}[f(Y)])g^U \mathbb{1}_{f(Y) > \mathbb{E}[f(Y)]}] \\
&\leq \mathbb{E}[(f(Y) - \mathbb{E}[f(Y)])g^L \mathbb{1}_{f(Y) \leq \mathbb{E}[f(Y)]}] + \mathbb{E}[(f(Y) - \mathbb{E}[f(Y)])g^L \mathbb{1}_{f(Y) > \mathbb{E}[f(Y)]}] \\
&= 0,
\end{aligned}
$$

as desired.  $\square$

**Lemma A.5** *[Hoeffding's Lemma [18]] Let $X$ be a mean 0 random variable such that $X \in [a, b]$ almost surely. Then, for all $\lambda \in \mathbb{R}$*

$$
\mathbb{E}[\exp(\lambda X)] \leq \exp\left(\lambda^2 \frac{(b-a)^2}{8}\right).
$$