# OpenReview forum: "Adaptive Conformal Inference Under Distribution Shift"
_NeurIPS.cc/2021/Conference — NeurIPS 2021 Oral_

### Official Review · Reviewer_2Sne · 2021-07-12

**Rating:** 8
**Confidence:** 4

**Summary:**

In this paper, the authors introduce an extension to (split) conformal prediction under distributional shift over time known as adaptive conformal inference (ACI). Instead of relying on exchangeability, the method relies on tracking a scalar parameter $\alpha_t$ which compensates for the discrepancy between the empirical miscoverage and target miscoverage $\alpha$. The authors provide theoretical coverage guarantees on long-run error rates under weak assumptions, and further provide concentration inequalities and single time coverage guarantees under stronger Markov conditions. The adaptive method is compared to usual conformal prediction on two examples.


**Limitations And Societal Impact:**

The authors have described the limitations of their theory, e.g. having a fixed $\hat{Q}$ with time.

**Main Review:**

Strengths:

I really enjoyed reading this paper. The writing is very clear, and the authors provide clear intuition for the theoretical results.  Although the method is simple, relying on adapting the miscoverage level $\alpha_t$ in an online manner instead of exchangeability is novel. The method remains very general like regular conformal prediction, requiring only a predictor. The generality of the theoretical results is also impressive, as the assumptions are quite weak for Section 4.1 and the Markov setup of Section 4.2 is sufficiently complex. The Propositions/Theorems are also easy to interpret.  The experiments convincingly demonstrate the correct coverage achieved by ACI under time-varying distribution shift, and the role of the sequence $\alpha_t$ is easy to interpret. The empirical results match nicely with theory in terms of long-run coverage.

#####################################################################

Weaknesses/Questions:

I only have minor suggestions:

1.) In the discussion, it may be worth including a brief discussion on the empirical motivation for a time-varying $\hat{Q}_t$ and $S_t$, as opposed to a fixed one as in Section 4.2. For example, what is the effect on the volatility of $\alpha_t$ and also on the average lengths of the predictive intervals when we let $\hat{Q}_t$ and $S_t$ vary with time?

2.) I found the definition of the quantile a little confusing, an extra pair of brackets around the term
\begin{equation}
\left(\frac{1}{|\mathcal{D}|} \sum_{\left( X_r,Y_r\right)\in \mathcal{D}} \mathbf{1}_{S(X_r,Y_r) \leq s}\right)
\end{equation}
might help, or maybe defining the bracketed term separately if space allows.

3.) I think there are typos in Lines 93, 136, 181 (and maybe in the Appendix too): should it be $\hat{Q}_t(1-\alpha_t)$ instead?


#####################################################################

Overall:

This is a very interesting extension to conformal prediction that no longer relies on exchangeability but is still general, which will hopefully lead to future work that guarantees coverage under weak assumptions. I believe the generality also makes this method useful in practice.


**Time Spent Reviewing:**

5 hours

---

> ### Author Response · Authors · 2021-08-10
> **Response to reviewer 2Sne**
>
> We thank the reviewer for their comments and are pleased that they enjoyed our paper. In response to the reviewer's specific comments:
>
> * In our opinion the benefits of fitting $S_t$ and $Q_t$ online are two-fold. First, using a good conformity score will give smaller prediction intervals. For a concrete example, suppose that $S_t = |\mu_t - Y_t|$ where $\mu_t$ is our estimate of the mean of $Y_t$ at time $t$. Using split conformal inference this conformity score will lead to intervals of the form $[\mu_t - \delta, \mu_t + \delta]$ for some parameter $\delta > 0$. The question at hand then is how large should $\delta$ be? To answer this, assume for simplicity that $Y_t \sim N(m_t,1)$. Introduce the notation $z_{\beta}$ to denote the $\beta$ quantile of $N(0,1)$. If $\mu_t =  m_t$ (i.e. the estimate is perfect) then $\delta \cong 1.96$ will lead to a valid $95\\%$ prediction interval. On the other hand, if $\mu_t = m_t+1$ (i.e. $\mu_t$ is a poor prediction), then we need to take $\delta \cong 2.6$, which leads to a larger interval. Following this principle, we believe that updating $S_t$ online should lead to more accurate point-predictions and thus smaller intervals. This idea has been investigated extensively in previous works on conformal inference (see for example the experiments in https://arxiv.org/pdf/1905.03222.pdf).
>
> The second benefit of fitting $S_t$ and $Q_t$ online is to dampen the distribution shift. In particular, when using our method it is highly beneficial to use a function $S_t$ with the property that $S_t(X_t,Y_t)$ is almost stationary. The choice of $S_t$ in section 2.2 tries to accomplish this by dividing by the variance estimate $\hat{\sigma}_t^2$. In fact, if the Garch(1,1) model was correct and  $\hat{\sigma}_t^2 = \sigma^2_t$ was a perfect estimate of the variance, then $S_t$ would be stationary. From the perspective of our algorithm, the distribution shift can be quantified by the change in the $1-\alpha$ quantile of $S_t$. If $S_t$ is stationary, then this quantile is stationary and there is zero distribution shift. Having low distribution shift will improve the coverage properties of our algorithm (see Theorems 4.1 and 4.2).
>
> The ideas discussed above are not fully explained in the current version of the paper. We will prepare a revision of the manuscript that more explicitly highlights these concepts.
> * We will add the additional brackets.
> * The reviewer is correct that there are typos appearing on lines 93, 136, and 181. We apologize for these mistakes and will correct them in our revision.

---

> > ### Comment · Reviewer_2Sne · 2021-08-23
> > **Response to Rebuttal**
> >
> > Thank you for the detailed answers. The above explanation on time-adaptive $S_t,Q_t$ is clear, and this discussion in the main paper would help make the paper more holistic. As pointed out by the other reviewers, an empirical comparison of the average interval widths between the adaptive and non-adaptive conformal method would also be very helpful for practitioners, maybe for the experiment in Section 5.

---

### Official Review · Reviewer_uBLA · 2021-07-14

**Rating:** 8
**Confidence:** 4

**Summary:**

This paper proposes a method for online adaptive conformal prediction, where the data generating distribution is unknown, and varies over time. Conformal prediction is a distribution-free uncertainty quantification methodology that forms prediction sets $\hat{C}$ for an input $X$ such that the response variable $Y$ is covered with probability $1 - \epsilon$. (This is similar in spirit to confidence intervals.) The foundational assumption supporting the majority of conformal inference techniques is exchangeability of calibration data and test data. This work departs from that limiting assumption, and develops a simple but rigorous approach that models continual distribution shift over time.

**Limitations And Societal Impact:**

I do not forsee any negative societal impact (other than that inherited by the blackbox models the conformal inference package wraps).

**Main Review:**

=== Strengths ===

- The paper is well-written, clear, and well-motivated. Distribution drift can be a significant issue in many real problems, and therefore adaptive confidence algorithms that can account for this are necessary.

- The main algorithm of the paper is simple and easy to implement (which is a strength!). It reminds me of proportional control for PID controllers, though the analysis and applications differ.

- The experiments are interesting and clearly real world tasks, that also clearly have distribution shift. This is great for illustrating to the community examples of problems the proposed approach helps solve. Empirically, the adaptive technique here performs well (especially compared to non-adaptive baselines that can fail catastrophically).

- The extended analysis in the hidden Markov model in Section 4.2 is helpful in illustrating what sort of performance we might be able to expect on a more precise level (as the guarantee in Section 4.1 is somewhat loose).

=== Weaknesses/Limitations ===

- The main drawback is that without further assumptions (such as those modeled in 4.2), we only have asymptotic control of the average error. In intermediate time intervals, we might have very conservative or anti-conservative predictions. That said, the theory in Section 4.2 is helpful in giving some intuition on these aspects, combined with the positive empirical results in Section 5.

- Returning to the control theory view, I wonder if it would be helpful to incorporate full PID control to achieve better dynamics (i.e., more quickly adjusting to sudden, but permanent shifts in distribution, or alternatively, dampening out adjustments due to outlier disturbances).

- The assumptions that this paper makes are minimal, which makes the possible approaches fairly limited. Though I agree that it would be out of the current scope, it would be interesting to see if some techniques from other recent distribution-shift conformal inference (e.g., Tibshirani et. al. and Cauchois et. al.) could be incorporated as part of the wrapped model to improve empirical performance. For example, if *batches* were observed at time t so that a flawed but decent likelihood ratio model could be fit (maybe with regularization on importance weights), perhaps the importance weighting technique of Tibshirani could be approximately used to improve estimation of $\hat{Q}_t$.

=== Justification for score ===

I think that this paper would make a good contribution to the NeurIPS program.

=== Minor comments ===
- typo in the proof of Lemma A.2 (double note denote).

**Time Spent Reviewing:**

7

---

> ### Author Response · Authors · 2021-08-10
> **Response to reviewer uBLA**
>
> We thank the reviewer for their time spent reviewing the
> manuscript. The reviewer notes three main limitations of the current
> work. The first drawback highlighted is that we only have asymptotic
> control of the error rate. We agree with the reviewer that this is a
> limitation of our work, but we are also of the opinion that no
> practical algorithm can hope to do better without additional
> assumptions on the size of the distribution shift. Namely, if the
> shift at a specific time-point is allowed to be arbitrarily large in
> magnitude, then no algorithm can hope to maintain coverage
> at that time. As the reviewer points out, our results in Section 4.2
> indicate how the asymptotic guarantees can be improved under
> additional assumptions.
>
> The reviewer notes two additional weaknesses/limitations of our work. In particular, they inquire about the relationship to PID control theory and they ask about a setting in which batches of i.i.d. data are observed. We agree that both these questions are quite interesting and are worthwhile directions for future work. For example, we are currently investigating whether ideas from PID control can assist in choosing the step-size $\gamma$ in a more principled manner. However, we also agree with the reviewer that this is outside the scope of the current article.

---

> > ### Comment · Reviewer_uBLA · 2021-09-03
> > **Thanks**
> >
> > Dear authors, thank you for the response. I look forward to the results of your current investigations in future work.

---

### Official Review · Reviewer_vCi2 · 2021-07-16

**Rating:** 6
**Confidence:** 3

**Summary:**

This paper provides a method for calibrating prediction sets in an online setting. The method adjusts the level of an underlying conformal prediction algorithm according to the number of miscoverage events observed previously in the trajectory. The authors prove that the empirical miscoverage rate never exceeds the desired level, regardless of the data-generating distribution at each step. The authors also provide an additional analysis of the tails of the empirical empirical error count in the setting where the data-generating distribution is a hidden Markov model (confirming that several intuitions apply in this case), as well as two applications involving real data.

**Limitations And Societal Impact:**

The authors include an interesting and broad discussion of the limitations of their method. They do not discuss potential negative societal impact, but their justification for not doing so in the Checklist seems reasonable.

**Main Review:**

This paper is cleanly written, and easy and enjoyable to read. The idea is interesting, and the strong guarantee provided by Proposition 4.1 in general settings is very nice. The results in section 4.2 nicely confirm several intuitions presented earlier in the paper.  Finally, the two real-world examples seem worthy and interesting applications, and additionally show some interesting failure modes of existing conformal prediction methods.

The empirical results provided, while thought provoking, could be improved by providing some details as to what the actual prediction intervals returned by the method look like (as opposed to just the miscoverage rates). This seems crucial given the generality of the method, which Proposition 4.1 shows would be correct (i.e. would produce the specified miscoverage rates) even if $\hat{Q}_t$ were a very poor approximation to the true quantile function.

Separately, while overall the paper is extremely well-written, I found the final paragraph of section 4.1 a little confusing. At a low level, my own calculations suggest the left-hand side of the equation under line 192 should be
$$
\mathbb{E}[\mathrm{err}_t] - \alpha
$$
instead of what is written, and the right-hand side of the equation under line 193 should
be
$$
\frac{1 - (1 - \gamma)^T}{T \gamma}|\alpha_1 - \alpha|
$$
instead of what is written (unless I have made a mistake).  I then do see an underlying similarity between the resulting bound and the right-hand side of (5), but I didn't understand the relationship between $\mathrm{max}\{\alpha_1, 1 - \alpha_1\}$ and $|\alpha_1 - \alpha^\ast|$ described (although I agree that the latter is an upper bound to it).

Finally, minor typos:

* Line 48 -- "points predictions"
* Equation under line 181 -- $\infty$ should be $-\infty

**Time Spent Reviewing:**

4

---

> ### Author Response · Authors · 2021-08-10
> **Response to reviewer vCi2**
>
> We are pleased that the reviewer found the paper enjoyable to read and we thank them for their comments. As discussed in response to other reviewers (SBVj, 2SNe) we will formulate a revised version of the manuscript that includes additional information and experiments demonstrating how the quality of the estimates of $S_t$ and $\hat{Q}_t$ impact the interval widths. In addition, the reviewer points out a few typos appearing in the manuscript. We agree with the reviewer's calculations and we will revise the manuscript to correct these mistakes.

---

> > ### Comment · Reviewer_vCi2 · 2021-08-22
> > **Response**
> >
> > Thanks for your response. The additional information regarding the interval widths you mention will I think be a valuable addition to the paper. I am still unclear on the discussion at the end of Section 4.1. While I see a similarity between the corrected formula and equation (5), I am still not completely clear on the argument being made here, and suggest clarifying this in the next version of the paper also.

---

### Official Review · Reviewer_apoZ · 2021-07-16

**Rating:** 8
**Confidence:** 3

**Summary:**

The paper proposes a conformal inference based technique that is an online adaptation to handle distribution shifts over time that has statistical guarantees for coverage of predictive distributions. The technique discretizes the updates and adapts marginal coverage with respect to limitations observed in prior steps. The authors demonstrate its usage in predicting market volatility and 2020 presidential elections.

**Ethical Concerns:**

No ethical concerns

**Limitations And Societal Impact:**

The authors discuss the limitations in the manuscript. I also provided more suggestions above to further enhance this discussion.

**Main Review:**

Originality: The authors propose an online adaptation of conformal inference to handle distribution shifts. There have been work on this front by Tibrishani  et al. and Cauchois et al. which were cited but I found the comparison of the proposed technique with respect to these techniques limited. The novelty on top of conformal quantile regression can be argued to be minimal and the authors propose a simple modification of these techniques to propose an online version. This modification is a well-known adaption of various techniques in general.

Quality: The paper provides theoretical analysis of the claims; however, I think there is a bit of a room for improvement in experimental results through comparisons with prior work.

Clarity: The paper is very well written and organized.

Significance: Online extension of conformal inference provides an interesting research direction. However, the models used in practice are generally frequently trained. Applying the adaptive version proposed in this paper on frequently trained models can still be applicable but the paper is lacking the experiments that provides comparisons to illustrate the added benefit of applying adaptive conformal inference in such scenarios. I think these additions would further validate the significance of the proposed approach.

Specific comments:
-	I think comparisons with respect to online trained models would be a highly important comparison to support the general usability of this technique.
-	I would recommend comparisons with respect to prior work that is already cited. For instance for the experiments with election results, the assumption of conditional distribution being constant between training and test sets by Tibrishani et al. is a valid assumption and it would have been interesting to see the predictions in comparison.
-	The proposed method is effective if the distribution shift happens gradually so that it is able to maintain it’s coverage over the shift period. The method will provide coverage guarantees when this change does not happen abruptly. This might be an important limitation to mention in the paper and discuss extensions.
-	The paper focuses on marginal coverage but conditional coverage (https://arxiv.org/pdf/2006.02544.pdf) is an important metric to illustrate statistical properties. If shown to have nice properties, this might make the results more powerful. I would recommend adding results with conditional coverage as well which will make the results more compelling.
-	Please comment on the jump from 84% to 95% local coverage level for Figure 2a.


**Time Spent Reviewing:**

5 hours

---

> ### Author Response · Authors · 2021-08-10
> **Response to reviewer apoZ**
>
> We thank the reviewer for taking the time to carefully review our manuscript. In response to each of the reviewers specific comments:
>
> * We are somewhat confused by the reviewer's request for experiments with "online trained models". In the experiments in Sections 2.2 and 5 the regression model and the quantile function are both fit in an online fashion. Thus, our experiments already consider online trained models. Perhaps the reviewer could clarify his/her request?
> * We agree with the reviewer that the assumption of Tibshirani et
>   al. may be valid in this setting. However, we do not see a natural
>   way to apply the method of Tibshirani et al. to the present
>   problem. Namely, the method of Tibshirani et al. requires us to
>   estimate the likelihood ratio for $X$ between the training and testing
>   distributions. In the current setting, it is possible that each data
>   point $(X_t,Y_t)$ comes from a unique distribution. Thus, we do not
>   see a natural way to compute this likelihood ratio. An alternative
>   method for exploiting the covariate information is to use $X_t$ to
>   help the estimation of $\alpha_t$. While we believe that this is an
>   exciting direction for future work, this is outside the scope of the
>   current article.
> * We only partially agree with the reviewer's comment that adaptive conformal inference only protects against gradual shifts. We agree with the reviewer that if the environment undergoes a large distrbution shift, then there will be a small period of time after the shift occurs in which the algorithm performs poorly. However, after this small period the method will re-calibrate to the new distribution. In our experiments adaptive conformal inference does a good job of handling large distribution shifts. See for example the bottom middle panel of Figure 1 where adaptive conformal performs well during the 2008 financial crisis. Furthermore, large distribution shifts are also accounted for in our theoretical results. Namely, note that the bound in Theorem 4.2 is controlled by the term $\mathbb{E}[|\alpha^*_{A_{t+1}} - \alpha^*_{A_{t}}|]$. This term can be small for two reasons: 1) if $A_{t+1} \neq A_t$, but $|\alpha^*_{A_{t+1}} - \alpha^*_{A_{t}}|$ is small, i.e. the shift is gradual and 2) If $\mathbb{P}(A_{t+1} = A_{t})$ is large, i.e. shifts are larger, but rare.  In our revision to the manuscript we will work to more clearly explain the behaviour of adaptive conformal in response to large and small shifts.
> * In the conformal inference literature approximate conditional coverage is usually achieved through the use of good conformity scores. Thus, the best way to achieve approximate conditional converge with our method is by using a good model of the environment and thus a good conformity score. Since this is the focus of a large body of other research (e.g. the article cited by the reviewer), we do not consider it to be within the scope of this paper.
> * In Figure 2a the counties appear in a uniformly random order. Thus, the baseline non-adaptive method shown in red achieves a coverage guarantee of $\mathbb{P}(Y_t \in \hat{C}_t) = 1-\alpha + \epsilon$ (for some negliglble error $\epsilon < 1/250$) for all $t$. Thus, the jump from 84\% to 95\% should simply be viewed as the natural variation of a mean 0.9 random walk.

---

> > ### Comment · Reviewer_apoZ · 2021-09-01
> > **Thank you authors for the clarifications**
> >
> > I thank the authors for their detailed response. I did not realized the model in Section 5 was fit in an online fashion and that resolves my comment. Their clarifications on this point and others resolved all of my concerns. Due to this, I am raising my score to 8.

---

### Official Review · Reviewer_SBVj · 2021-07-16

**Rating:** 5
**Confidence:** 3

**Summary:**

This paper proposed an adaptive method to estimate a confidence set such that the target value falls into (conformal with) the set within a given probability. It adjusts the coverage level \alpha_t in the current round t based on the error from previous rounds (see Eq.(2)). Empirical demonstrations on stock markets and the 2020 presidential election poll show that the proposed adaptive conformal inference (ACI) method can better reach the targeted coverage level than using a fixed alpha.

**Limitations And Societal Impact:**

The authors have adequately addressed the limitations and potential negative societal impact.

**Main Review:**

Pros
- A straightforward method that is well-explained
- Theoretical justification under certain assumptions

Cons
- The theoretical analysis does not lead to practical guidance
- Insufficient discussion and comparison with related papers

Detailed comments

Overall, it is an interesting paper demonstrating the possibility and advantage of using an adaptive method to handle distribution shifts in conformal inference. However, I have the following concerns.

1. Regarding the theoretical analysis, it lacks practical guidance on choosing the step-size gamma. More specifically, the step size in Thm.4.2 relies on a future quantity that is unknown in practice. Given that the Markov chain is assumed to be mixed, wouldn't it be possible to derive a more practical choice of gamma based on the history?

2. The current paper discussed the coverage but lacks a discussion on the interval width, which is also important in practice. For example, see Kivaranovic et al. (2020) and Chen et al. (2021), which provide more thorough treatment for the problem and it would be necessary to discuss/compare.

- Kivaranovic, D., Johnson, K.D. and Leeb, H., 2020, June. Adaptive, distribution-free prediction intervals for deep networks. In International Conference on Artificial Intelligence and Statistics (pp. 4346-4356). PMLR.
- Chen, H., Huang, Z., Lam, H., Qian, H. and Zhang, H., 2021, March. Learning prediction intervals for regression: Generalization and calibration. In International Conference on Artificial Intelligence and Statistics (pp. 820-828). PMLR.

3. Empirical studies.

- L145 mentioned that "These stocks were selected out of a total of the 13 stocks that we examined because they showed a clear failure of the non-adaptive method". However, it seems to me the non-adaptive alpha works fine for AMD. If ACI is indeed performing well for the rest out of 13 stocks, it would be helpful to see the actual improvement (maybe in Appendix) instead of simply claim it without evidence.

- For the results in Fig.2, it would be informative to see the deviation from the ideal 0.9 as sigma changes for the two methods. Currently, only specific values are chosen and it is not clear why.

- There is no comparison to competitive methods in the literature other than a fixed-alpha baseline. It would be more convincing if the proposed method can be compared to the methods mentioned in the above reference.

Minors

- Appendix A.2, it might be better to put both curves (using (2) and (3) respectively) in one figure so that they can be compared more directly.
- The alpha in the GARCH(1,1) model (Equation below L127; also the \hat{\alpha}_t in L132) has a different meaning from the alpha in most parts of the paper (E.g., in L137), which is likely to confuse readers.
- L294, sigma>0 should be sigma>=0 as seen in the leftmost of Fig.2.

**Time Spent Reviewing:**

5

---

> ### Author Response · Authors · 2021-08-10
> **Response to reviewer SBVj**
>
> We thank the reviewer for their thoughtful comments. In response to each of the reviewers specific points:
>
> 1. We agree that this is one of the main limitations of the current work and that in principle choosing $\gamma$ adaptively based on the history should be feasible. We consider this to be an exciting direction for future work. As our experiments show, even a simple choice of $\gamma = 0.005$ can be quite successful in practice across many different datasets.
> 2. Algorithms for achieving small interval lengths have been heavily investigated in prior works. Since the methodology used in this paper can be applied in concert with any method for fitting the quantile function $\hat{Q}_t$, it is certainly possible to combine our method with previously developed approaches for minimizing the interval width (e.g. the approach of Chen et al.). As the focus   of our work is on handling the distribution shift and interval length has been well studied in other articles, we ommited it from our investigation.  With that said, we agree with reviewer 2Sne that an investigation of how fitting $S_t$ and $\hat{Q}_t$ online impacts interval length is highly relevant to our method. We will revise the manuscript to include additional experiments that investigate how the quality of the estimates $S_t$ and $\hat{Q}_t$ impact the lengths. We will also include the two references suggested by the reviewer. This will help guide readers unfamiliar with prior work in this area to methods for minimizing interval width.
> 3.
> * We will add plots for the additional stocks to a revised version of the manuscript.
> * As noted in the caption to Figure 2, we choose to display these four values for $\sigma$ because they capture the change in the observed coverage frequency as $\sigma$ ranges from 0 to $\infty$.  Other values of $\sigma$ in between those shown will give curves that are intermediate to those already displayed.
> * We are not aware of any competing method in the literature that directly addresses the problem investigated in this article. The closest methods that we are aware of are those discussed in Section 3, but these methods are designed for cases where there is only a single training  distribution and a single testing distribution and thus are not directly applicable to the current setting.
>
> Additionally, we thank the reviewer for their minor comments. We will incorporate the changes suggested by the reviewer in our revision.

---

### Official Review · Reviewer_jzW6 · 2021-07-19

**Rating:** 7
**Confidence:** 4

**Summary:**

This paper develops an online predictive inference technique motivated by conformal inference. The proposed method achieves the target coverage rate ($1-\alpha$) as $T \to \infty$. For finite $T$, the coverage is at most $O(1/T)$ away from $1-\alpha$. Experiments on two real datasets illustrate that the method exhibits improved behavior as compared to a non-adaptive baseline.

I feel that the main insight/contribution of the paper is to draw an elegant connection between two apparently disjoint areas, namely 1) post-hoc uncertainty quantification and 2) online learning. The methodology of the paper is mostly inspired from a post-hoc uncertainty quantification setup inherent in split conformal, where some part of the model is treated as 'fixed' and only a few parameters are optimized towards some goal (in this case, a single parameter, the quantile level $\alpha_t$ is learnt). On the other hand, the theoretical analysis is very different from other conformal papers. It is in the adversarial setting and the technical tools used are similar to those used in the online learning literature.



**Limitations And Societal Impact:**

To the best of my understanding, the approach proposed does not guarantee that C_T(\alpha_T) has 1-\alpha coverage for a fixed time T, no matter how large T is. Proposition 4.1 only holds on _average_ across time. The authors make the following claim in the abstract, "our adaptive approach provably achieves the desired long-term coverage frequency irrespective of the true data generating process". I think that this claim needs to be (transparently) supplemented with the limitation described above.

**Main Review:**

As mentioned above, I think that the work is an original and significant contribution to the conformal literature. The experiments, although not extensive, nicely supplement the theory. Being quite familiar with conformal inference, I found the paper easy to follow (except the theorems in Section 4.2; see more below). However, I felt that the paper lacked a certain 'attention to detail' or 'polish' that the technical ideas are highly worthy of. I have made some suggestions below.

_Major comments regarding writing:_
- Line 30: This is not true for many recently popular conformal techniques: jackknife+ [1], cross-conformal [2], and out-of-bag conformal [3], all of which (roughly) get $1 - 2\alpha$ coverage. Please update or qualify this claim.
- Section 4.2: I was unable to really understand Theorems 4.1 and 4.2 due to the little t's which are not qualified as part of the statement. I am not sure what the expectation is over in defining $B$ and $\sigma_B$. Similarly I am not sure what the expectation is over on the LHS and RHS of eq. (7). Could the authors please clarify this?
- Line 279: Since Neurips is an international conference, I believe it should be specified that the authors mean the 2020 United States of America presidential election. The description of the dataset (until line 291) strongly assumes that the reader is familiar with the US election system: Who is Joe Biden? What is election night? The west (east) coast of what?

_Minor/subjective comments:_
- I felt that the paper was heavy on notation, which could be overwhelming for readers unfamiliar with conformal inference. The authors can consider reducing notation in order to reach a larger audience. I also propose a different way of looking at the algorithm proposed by the authors in the next section, which would probably require less notation.
- Section 2.2: I felt that this section broke the flow of the rest of the paper
- Title of Section 4.1: what is the meaning of 'general results'?
- Line 221: “there is little doubt that the theory can be extended, recall our main goal [is] to get useful and simple results”: Did the author mean to say that it is unlikely that the theory can be extended to practical settings? What they have said conveys the opposite meaning (there is no doubt that the theory can be extended to practical settings).
- Initially the subscript to M denotes ‘time’. Later, in Theorem 4.2, it denotes ‘action’. I suggest using slightly different notation (such as bracketing) for the two.

**_A reinterpretation of the methodology proposed in the paper_**:

I had a reinterpretation of the method proposed by the authors, based on 3 technical tools: pinball loss [4], online subgradient descent [5], and nested conformal [3].

First, for any time i, define a family of nested sets ${F_t}$ based on [3]; such a family can be defined for any nonconformity score. Then define $ t_i := \inf t: Y_i \in F_t(X_i)$. Suppose $F_{s_i}$ is the predicted prediction set at time step i, then for $1-\alpha$ coverage, what is needed is that $s_i \geq t_i$, at least $1-\alpha$ of the times. In other words, for optimal coverage, **$s_i$ should be the $1-\alpha$ quantile of $t_i$**. A popular way to optimize for quantiles is to use the pinball loss. The proposed method (2) optimizes $\sum_{i=1}^T \text{pinball-loss}(1-\alpha,s_i, t_i)$ using online subgradient descent.

Do the authors agree with this interpretation? If yes, including it would enhance the paper, maybe even in an Appendix. I personally found this to be a cleaner way to understand the main idea: it requires less notation and alludes to other ideas that readers may already be familiar with.

_References:_

[1] https://arxiv.org/abs/1905.02928

[2] http://proceedings.mlr.press/v91/vovk18a.html

[3] https://arxiv.org/abs/1910.10562

[4] http://www.econ.uiuc.edu/~roger/research/rq/QRJEP.pdf

[5] https://www.jmlr.org/papers/volume12/duchi11a/duchi11a.pdf


**Time Spent Reviewing:**

8

---

> ### Author Response · Authors · 2021-08-10
> **Response to reviewer jzW6**
>
> We thank the reviewer for their positive comments. We believe that all of the reviewers comments can be addressed in a minor revision to the manuscript. More specifically we will make the following changes.
>
> *Response to major comments*
> * Our comment on line 30 refers specifically to the original work of Vovk and colleagues (as in e.g. https://link.springer.com/book/10.1007/b106715). We will update line 30 to make this clear.
> * Because we are working with the stationary distribution of $(\alpha_t,A_t)$ the choice of little $t$ is arbitrary. We will update the writing around Theorems 4.1 and 4.2 to make this clear. On the left-hand side of equation (7) the expectation is over the pair $(A_t,\alpha_t)$ drawn from its stationary distribution. Similarly, on the right-hand side the expectation is over the pair $(A_{t},A_{t+1})$, again drawn from its stationary distribution. We will update the notation to make this clear.
> * We will update the writing in section 5 to make it more approachable by a non-US based audience.
>
>
> *Response to minor comments*
> * The reviewer suggests a new interpretation of our method that frames our algorithm as a specific form of online subgradient descent. We agree that this interpretation is quite interesting and that it may be easier to understand for readers who are already familiar with the online-learning literature. If space permits we will include a description of this interpretation in the revised version.
> * We believe that the ability of our method to handle real-world shifts is a key strength of our work. Thus, we think it is good to have the results of Section 2.2 appear early in the article to highlight this fact.
> * We use the term "general results" to refer to results that do not require any assumptions on the data-generating distribution. To clarify this we will rename this section "Distribution-Free Results".
> * On line 221 we wish to convey that we are confident that our results can be extended to more general settings (e.g. $\hat{Q}_t$ not fixed, other models for the data-generating distribution). We believe that this line conveys this as written.
> * We will change the notation from $M_{A_t}(\beta)$ to $M(\beta | A_t)$. We hope this will help alleviate the confusion and more clearly indicate that $M_{A_t}(\beta)$ refers to a conditional probability, conditional on $A_t$.
>
> *Limitations and Societal Impact*
>
> We will update the phrasing in the abstract to "Over long time intervals our adaptive approach provably achieves the desired average coverage frequency irrespective of the true data generating process."

---

> > ### Comment · Reviewer_jzW6 · 2021-08-24
> > **Follow-up**
> >
> > Dear Authors,
> >
> > Thank you for response. Technically, the paper stands out from other papers in conformal, and has a good chance of having significant long-term impact. But in terms of writing and exposition, I encourage you to polish the paper and the mathematical notation based on the reviews.

---

### Decision · Program_Chairs · 2021-09-27

**Decision:**

Accept (Oral)

**Comment:**

The paper proposes an adaptive conformal inference (ACI) that relaxes the classical exchangeability assumption by leveraging the online learning framework, and proposes a simple algorithm that achieves the target coverage rate as the number of steps approaches infinity. There is a consensus among the expert reviewers, with which I also concur, that the main idea of combining conformal inference with online learning is novel and interesting, and that the paper provides a fundamental contribution to the field of conformal inference. Hence, I recommend an acceptance.

One of the reviewers raised the concern regarding experimental results, which I consider a minor point. Nevertheless, improving the clarity of this paper, e.g., by making the paper accessible to broader audience, and providing thorough discussion with respect to the related work, as suggested by the reviewers, will further increase the impact of this work.

The authors provided a rebuttal which was also positively acknowledged by the reviewers.